# Observation of elastic topological states in soft materials

Shuaifeng Li[1,3], Degang Zhao[2,3], Hao Niu[1,3], Xuefeng Zhu[2,3] & Jianfeng Zang [1,3]

Topological elastic metamaterials offer insight into classic motion law and open up opportunities in quantum and classic information processing. Theoretical modeling and numerical simulation of elastic topological states have been reported, whereas the experimental observation remains relatively unexplored. Here we present an experimental observation and numerical simulation of tunable topological states in soft elastic metamaterials. The on-demand reversible switch in topological phase has been achieved by changing filling ratio, tension, and/or compression of the elastic metamaterials. By combining two elastic metamaterials with distinct topological invariants, we further demonstrate the formation and dynamic tunability of topological interface states by mechanical deformation, and the manipulation of elastic wave propagation. Moreover, we provide a topological phase diagram of elastic metamaterials under deformation. Our approach to dynamically control interface states in soft materials paves the way to various phononic systems involving thermal management and soft robotics requiring better use of energy.

[1] School of Optical and Electronic Information and Wuhan National Laboratory for Optoelectronics, Huazhong University of Science and Technology, Wuhan 430074, China. [2] School of Physics, Huazhong University of Science and Technology, Wuhan 430074, China. [3] Innovation Institute, Huazhong University of Science and Technology, Wuhan 430074, China. Correspondence and requests for materials should be addressed to J.Z. (email: jfzang@hust.edu.cn)

Topology describes the properties of space under continuous deformation in mathematics. The concept has been used to explain band structures in condensed matter physics, resulting in the theoretical predication and experimental observation of topological insulator in electronic system[1,2], and recently also in photonic[3–6] and phononic systems[7–13]. Topologically protected wave propagation possesses prominent applications in quantum computation and communication field due to its remarkable characteristic: the robust defect-immune transport[1]. Recently, significant research efforts devoted in phononic topological insulators provide a new way to manipulate sound propagation, such as vibration isolation and particle manipulation. Topological insulator has been successfully demonstrated in airborne acoustics by deliberate design of materials parameters, realizing backscattering-immune one-way airborne sound transport[10]. Besides, topological transport for phonon has been theoretically modeled and numerically simulated through constructing spring and mass system in discrete solids[12, 14–16]. Topologically protected helical edge state is further numerically realized in continuum solids[7]. However, it remains a challenge to experimentally realize elastic topological states in real materials, which limits its further applications on elastic devices, such as elastic energy storing[17] and elastic wave guiding technology[18].

Elastic waves, also known as small oscillations in solids, have potential applications in information carrying[19] as well as seismic monitoring[20]. Through creating bandgaps in architected materials with periodic porous structures, elastic waves can be dramatically attenuated, which is particularly useful in vibration isolation[21]. Distinct from acoustic waves and electromagnetic waves, elastic waves are complicated and hard to control because of their richer polarizations[22], while topological insulator opens a new avenue to control elastic waves. The backscattering-free nature of topological transport opens a possibility for large-scale phononic circuits[7]. Thanks to the development of advanced fabrication technique, such as directional solidification, elastically anisotropic and isotropic materials can be successfully fabricated now, generating peculiar elastic properties in selected directions. Such materials usually keep in a fixed structure and geometry after fabrication, resulting in fixed properties and functionalities. Whereas, soft material is capable of reversible mechanical deformation over its global and partial structure, providing a new degree of freedom to tune the properties or functionalities of the system. A variety of soft tunable acoustic devices have been reported. For example, the width and position of the phononic bandgap can be tuned through deforming elastomeric helix array[23] and/or buckling of elastomeric beam connected with local resonator[24]. Besides, the programmable mechanical behaviors have been achieved in a mechanical metamaterial, which may inspire new tunable devices[25]. Moreover, the intrinsic property of nonlinear mechanics in soft materials has enabled novel functions that do not exhibit in traditional elastic systems[26,27]. The tunable topological zero-energy motions based on Maxwell framework consisting of rods and hinges have been put forward. While they have the impact on novel machines and robots, the transport of elastic waves has not yet been directly revealed and the framework can hardly be considered as the continuous medium[28,29]. The combination of soft materials with high-frequency topological states offers unprecedented opportunities, which requires insight exploration.

Here we present an experimental observation and numerical simulation of tunable topological state in soft elastic metamaterials. The on-demand reversible switch of topological phase has been achieved by changing filling ratio, stretching, and/or compressing soft elastic metamaterials. We further demonstrate the dynamical tunability of topological state by mechanical deformation, including switch modulation and frequency modulation, as well as manipulation of elastic wave propagation. Moreover, we provide a topological phase diagram as a general scheme to design tunable topological states in soft elastic metamaterials. Our research provides a way to manipulate elastic waves artificially and opens an avenue to the development of soft topological insulator.

## Results

**Design of soft metamaterials.** Figure 1a presents the soft metamaterial with periodic honeycomb holes of air in a rectangle silicon rubber (Ecoflex) slab of 180 mm × 52 mm × 10 mm. The sample can be stretched or compressed resulting in the rearrangement of lattice and the reshaping of the air cylinder scatterers as illustrated in the inset of Fig. 1a. We select a hexagonal unit cell, which is shown in Fig. 1b. Through adjusting the nearest coupling, namely, manipulating $d/R$ to adjust the filling ratio, a twofold Dirac cone at M point can be formed by accidental degeneracy, as presented in Fig. 1c. The open and reopen of the Dirac cone can be controlled by continuously changing the filling ratio of the honeycomb lattice. When $d/R = 0.7156$, a Dirac cone appears at the edge of the Brillouin zone (M point). As $d/R$ is reduced to be 0.6 or increased to be 0.78, the Dirac cones are opened to be a bandgap along ΓM direction and all directions, respectively. Thus, as the increasing of the filling ratio, bandgap along ΓM direction exhibits the open, close, and reopen evolutionary process, as presented in Fig. 1c.

The controlled bandgap evolutionary process can also be achieved by stretching or compressing the soft material. When the soft elastic metamaterial (6 × 6 unit cells) is subjected to tension or compression strain, each hexagonal cluster (enclosed by six air pillars) and each individual air pillar undergo shape change. A two-dimensional (2D) model with plane strain condition is used to analyze the shape changes of the unit cell and the strain responses of the elastomer. Here the elastomer is delineated by a nearly incompressible Yeoh hyperelastic model. The strain-dependent shape changes are presented in Supplementary Fig. 1. When the elastic metamaterial is uniaxially stretched, the length of unit cell increases and the width of it decreases due to Poisson effect. Conversely, under the compressive strain, the length of unit cell decreases and the width of it increases. In both circumstances, circular air holes become to be elliptic. We set $d/R = 0.68$ and calculate band structures as a function of applied strain, as presented in Fig. 1d. When the system is under compression strain $\varepsilon = -4.44\%$, an absolute bandgap is observed between the first and the second bands. As the applied strain increases to $\varepsilon = -1.53\%$, a Dirac dispersion relation can be observed at M point. As the elastic system is stretched to $\varepsilon = 3.16\%$, the Dirac cone previously formed at M point disappears, shifting to MK direction, resulting in a bandgap along ΓM direction. It should be noted that the longitudinal wave bands are also exhibited in band structure (dashed lines in Fig. 1c, d). But in this work we only focus on the transverse modes. Here the applied strain on the elastic metamaterial is small enough so that the effect of stress on elastic wave propagation is negligible[30], as well as the effect of pesudomagnetic field due to the gradient of strain[31–33]. Therefore, we only consider the effect of the different geometry configurations of soft lattice on wave propagation.

To investigate the topological properties of this system and verify the fact of topological phase transition, we calculate the topological invariant, namely Zak phase of each band in ΓM direction, using symmetry analysis method[34,35]. Given the mirror symmetry of our physical system, the Zak phase is quantized and ensured to characterize the topology of the bulk. For each bulk band, the Zak phase should be $\pi$ if the eigenmode at center of Brillouin zone possesses different symmetries with that at edge.

Otherwise, the Zak phase is 0. As an example, Supplementary Fig. 2 presents eigen displacement field distributions of transverse modes at the band edges when $d/R = 0.6$ and 0.78 without applied strain. Similar analysis can also be carried out to determine the Zak phase in deformation process. Through analyzing the symmetry properties, we obtain the Zak phase of each band along ΓM direction, marked in Fig. 1c, d, exhibiting a distinct topological phase transition, which is also known as band inversion. We can similarly obtain the sgn (ς) of bandgap associated with Zak phase by a simple expression[36]:

$$\mathrm{sgn}\left(\varsigma^{(n)}\right) = (-1)^n (-1)^l e^{i\sum_{m=0}^{n-1}\theta_m^{Zak}} \qquad (1)$$

where $n$ is the sequence of bandgap and $l$ is the number of crossing points beneath this bandgap. The resultant bandgap signs are marked in Fig. 1c, d. The analysis details are seen in Supplementary note 1.

**Topological phase diagram**. The topological phase transition at M point has been achieved either by filling ratio or by mechanical deformation of elastic metamaterials. Further, we demonstrate the variation of topological properties as a function of $d/R$ or strain. As shown in Fig. 2a, the frequency range of the bandgap in ΓM direction decreases monotonically as the $d/R$ increases from 0.5 to 0.9, with a Dirac point formed at $d/R = 0.7156$. If we set the filling ratio of $d/R$ to be fixed values, e.g., $d/R = 0.6$, 0.68, or 0.78, we can get a diagram with three plots of bandgap variation, experiencing a similar topological phase transition in the strain range of $-8.89$ ~ 8.89%, as presented in Fig. 2b. Three topological transition points are observed at strain of $-5.71\%$, $-1.375\%$, and 2.66% for $d/R = 0.6$, 0.68, and 0.78, respectively. If we draw a dash-dotted line to connect all of the transition points together, we can divide the diagram into two regions, with the bandgap sign ς > 0 and <0, respectively. We call the dash-dotted line in Fig. 2b as a topological phase transition line. Since the bandgap sign is related to the reflection phase and further associated to the surface impedance

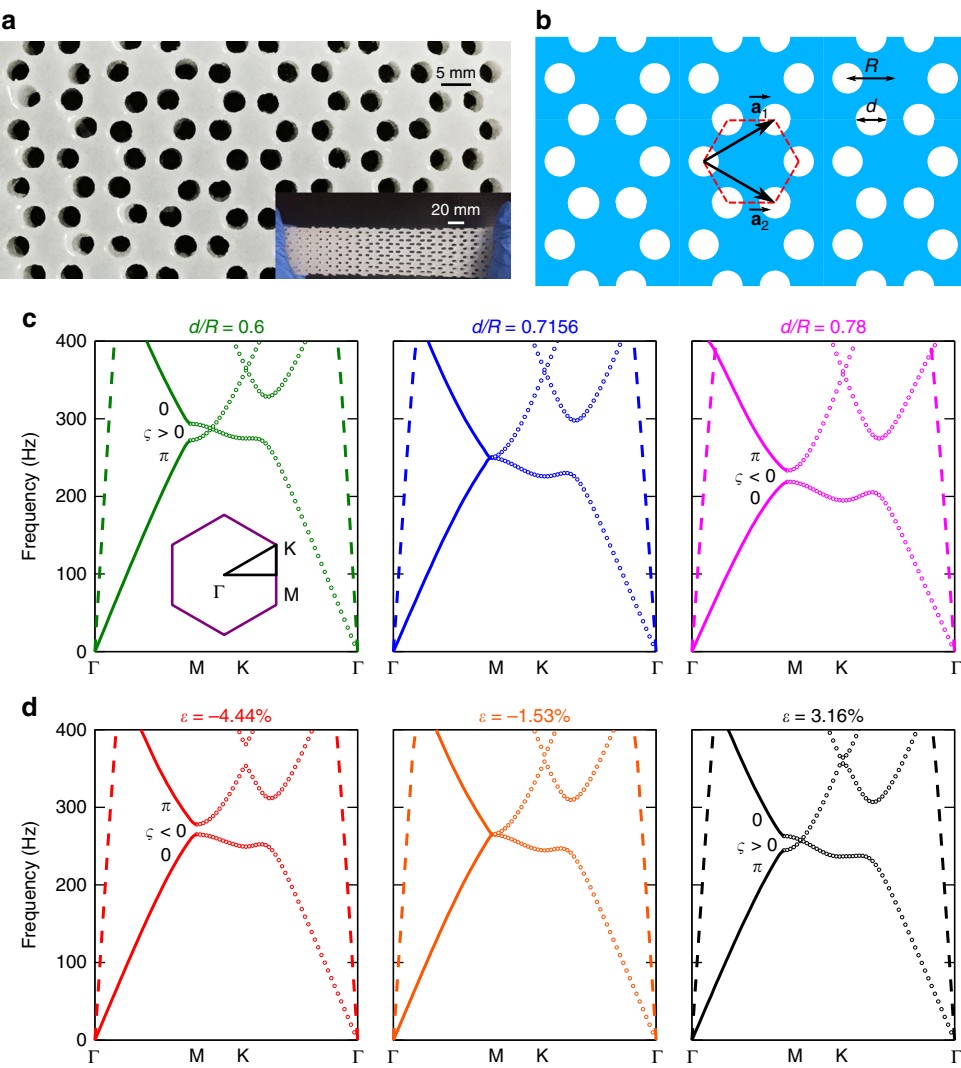

**Fig. 1** Design of topological elastic metamaterials and two band inversion processes. **a** Soft elastic metamaterial with periodic honeycomb holes of air in a rectangle silicon rubber (Ecoflex). The inset shows the stretchability of the soft metamaterial. **b** The schematic of our soft elastic metamaterials. Red dashed hexagon is the primitive cell with the hexagon edge length $R$ and hole's diameter $d$. $\vec{a_1}$ and $\vec{a_2}$ are the basic vectors. **c** Band inversion process as a function of filling ratio of $d/R$ without applied strain. The filling ratios of $d/R$ chosen are 0.6, 0.7156, and 0.78. Dashed lines indicate longitudinal wave bands. Solid lines and dotted lines indicate transverse wave bands in ΓM direction and other directions, respectively. Inset is the irreducible Brillouin zone. **d** Band inversion process as a function of strain with fixed filling ratio of $d/R = 0.68$. Three strains chosen from compression to tension are $-4.44$, $-1.53$, and 3.16%. The calculated Zak phase could be 0 or $\pi$, which is marked on the corresponding bulk band in ΓM direction (solid lines) in **c** and **d**. The calculated bandgap signs ς are marked in the corresponding bandgap. All band structures are from numerical simulation

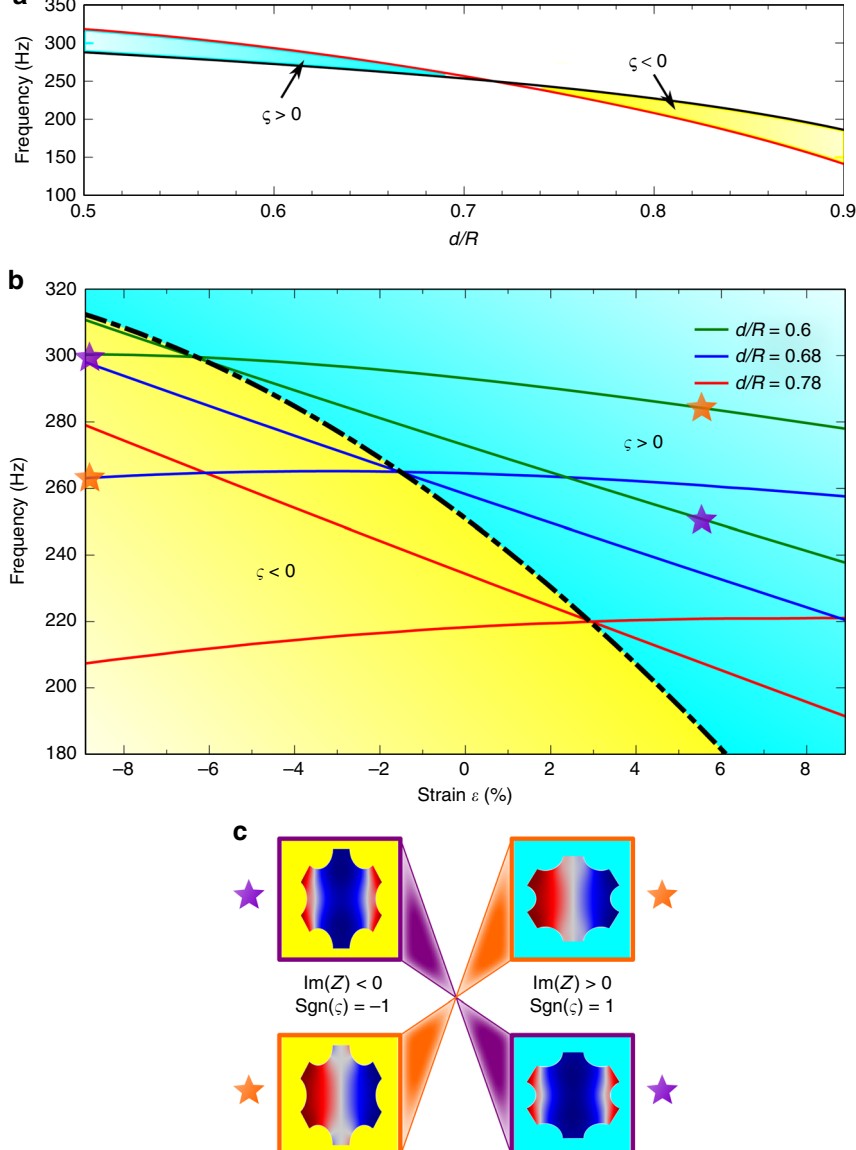

**Fig. 2** Topological phase diagram. **a** The frequencies of two states at M point as a function of filling ratio of $d/R$ without applied strain. The bandgap signs of the cyan and yellow regions are $\varsigma > 0$ and $<0$, respectively. **b** Topological phase diagram as a general design scheme consists of two domains separated by topological transition line (black dash-dotted line) that is formed by connecting the topological transition points of different filling ratio systems. Green, blue, and red solid lines are obtained by investigating the frequencies of two band-edge states at M point as a function of strain. The bandgap signs $\varsigma$ are marked in the yellow and cyan regions. **c** Real parts of eigen vertical vibration modes at M point inversion represents topological phase inversion in solid. The vibration modes correspond to the stars marked in **b**. The imaginary part of surface impedance and bandgap sign are marked between two vibration modes. All data are from numerical simulation

$Z(\omega, \mathbf{k}_{\parallel})$, the topological phase diagram can be considered as the surface impedance diagram as a function of strain. One region (yellow region) with $\varsigma < 0$ has $Z(\omega, \mathbf{k}_{\parallel}) < 0$ and the other region (cyan region) with $\varsigma > 0$ has $Z(\omega, \mathbf{k}_{\parallel}) > 0$. The real parts of eigen vertical displacement fields of the two distinct regions are shown in Fig. 2c, indicating topological phase transition with even and odd Bloch modes alternation.

The topological phase diagram shown in Fig. 2b provides a general scheme to design topological interface states in soft elastic metamaterials. In order to form topological interface states, the bandgaps of two metamaterials must share an overlapped frequency range with a band inversion. According to the phase diagram, we can construct a topological system by combining two metamaterials with different topological phases sharing the overlapped frequency range, denoted with ($d_1/R$, $\varepsilon_1 | d_2/R$, $\varepsilon_2$).

The possible constructions could be but not limited to (0.6, 5.56%|0.68, −8.89%) in the frequency range of 263 ~ 283 Hz, (0.6, 0%|0.68, −8.89%) in the frequency range of 272 ~ 293 Hz, and (0.6, −1.11%|0.68, −8.89%) in the frequency range of 277 ~ 295 Hz. The uniaxial strains we mentioned above are along horizontal direction. Actually, the topological phase can also be inverted by changing the strain along vertical direction. The phase diagram using the uniaxial strains along vertical direction is illustrated in Supplementary Fig. 3.

**Observation of topological interface state**. When two elastic systems with different topological invariants are edge-to-edge joint, the topological interface states can be predicted to emerge according to the surface impedance match condition $Z_L(\omega, \mathbf{k}_{\parallel}) +$

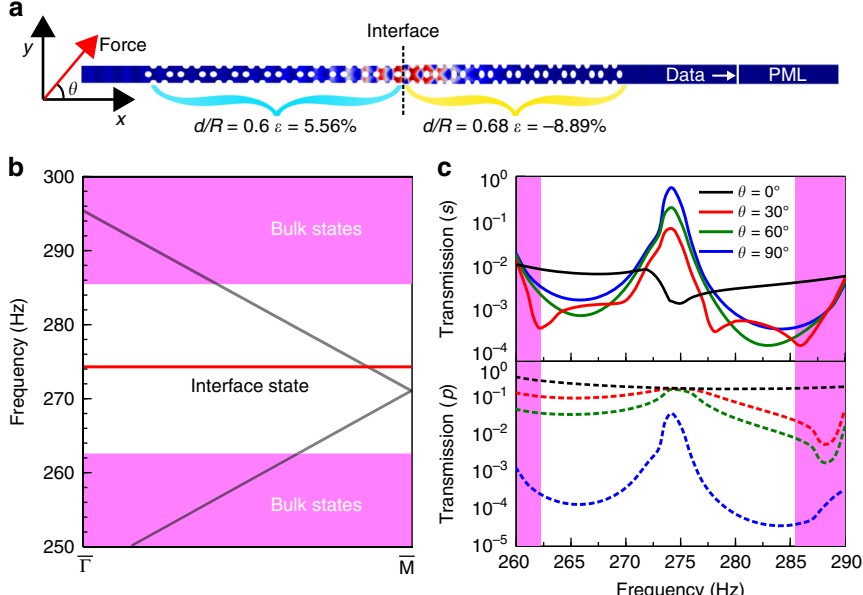

**Fig. 3** Numerical observation of topological interface states. **a** The numerical calculation uses a ribbon with periodic boundary condition, consisting with two elastic metamaterials, (0.6, 5.56%|0.68, −8.89%). The harmonic force with tunable angles is applied on the edge of the ribbon. **b** The simulated projected band structure along $\mathbf{k}_x$ direction indicated by $\overline{\Gamma M}$ with transverse interface modes. The bars above Γ and M are used to distinguish the Γ and M from band structure of unit cell. Red line indicates interface state independent of bulk state (magenta region) and gray lines indicate longitudinal wave modes. **c** The simulated transmission spectra as a function of excitation angle (θ = 0°, 30°, 60°, and 90°). The transmission spectra for transverse modes (marked by s) and longitudinal modes (marked by p) are presented for comparison

$Z_R(\omega, \mathbf{k}_{\parallel}) = 0$. As a typical example, we demonstrate the numerical observation of topological interface state by constructing a ribbon with two elastic metamaterials, (0.6, 5.56%| 0.68, −8.89%), which is presented in Fig. 3a. By combining two elastic metamaterials together, a $30 \times 1$ supercell is formed to calculate the projected band structure. As shown in Fig. 3b, a flat band of transverse mode at the frequency of 274 Hz within the bulk bandgap is observed in the projected band structure along $\mathbf{k}_x$ direction. The real-space distribution of displacement field is displayed in Fig. 3a with a transverse input excitation of 274 Hz applied on one side of the ribbon. We find that vibrations locate mainly at the interface between two elastic metamaterials and attenuate dramatically into the bulk, which coincides with the field distribution of the eigenmode in the flat topological band (Supplementary Fig. 4). Thus, the flat band is the topological interface mode independent from bulk modes. Figure 3c presents the transmission spectra as a function of different input excitation angles, with coexistence of transverse excitation and longitudinal excitation. As the excitation angle θ increases, the fraction of input transverse wave increases. No transmission peak of transverse wave is observed when θ is 0°. As the θ changes to 30°, 60°, and 90°, a transverse wave transmission peak emerges at the frequency 274 Hz and gradually grows to maximum. No evident peak is found in the longitudinal wave transmission spectra in this process, and the small peak at θ = 90° may be attributed to the conversion of longitudinal wave to transverse wave.

The topological interface states have been numerically achieved by combining two metamaterials with different filling ratios according to the phase diagram in Fig. 2b. Actually, we can even obtain the topological interface states by using two metamaterials with the same filling ratio if we consider both the phase diagram with horizontal strain in Fig. 2b and the phase diagram with vertical strain in Supplementary Fig. 3. For example, the projected band structure and eigenmode of the elastic system of filling ratio of $d/R = 0.68$ with strains in both horizontal direction and vertical direction are presented in Supplementary Fig. 5.

In order to experimentally observe the topological interface states, we apply the excitation force along the interface between two metamaterials, as presented in Fig. 4a. We calculate the projected band structure along $\mathbf{k}_y$ direction using the same supercell as calculation in Fig. 3b. As shown in Supplementary Fig. 6, a topological flat band (red dots) is observed near Γ point and a series of discrete modes are found above and below the flat band. It is interesting to note that when elastic waves with different frequencies enter the elastic metamaterial, the waves with frequencies above and below the flat band will be separated. The elastic wave with the frequency as same as that of flat band will be localized on the interface, exhibiting the topological interface state. While the elastic waves with lower frequency or higher frequency, they will propagate in right or left direction, respectively. This splitting propagation is clearly revealed by the vertical displacement field distribution at three typical frequencies, 261, 274, and 286 Hz, as shown in Fig. 4b.

We fabricate an elastic metamaterial sample comprised of air hole array (seen in Methods) with the setup of (0.6, 5.56%|0.68, −8.89%) as presented in Fig. 4a and Supplementary Fig. 9. A shaker to excite the elastic wave is placed on the interface of the sample and an accelerometer is placed in 24 holes one by one along the cyan line marked in Fig. 4b to detect the displacement. When the frequency of excitation signal is 274 Hz, the detected displacements at 24 holes are summarized in Fig. 4c. We find that the magnitude of displacement reaches maximum near the interface, and declines sharply away from interface. It is noted that the vibration at the left of the interface drops slower than that at the right, which is consistent with the simulation result in Fig. 4c and the field distribution in Fig. 4b. Note that inserting an accelerometer into the hole to measure the displacement will bring added mass to the sample. The accelerometer method has been employed as an effective way to detect vibration of elastic metamaterials[21,24]. Considering the stable characteristic of topological interface state, Majorana edge states have already been observed by using accelerometers[37]. In our case, the further

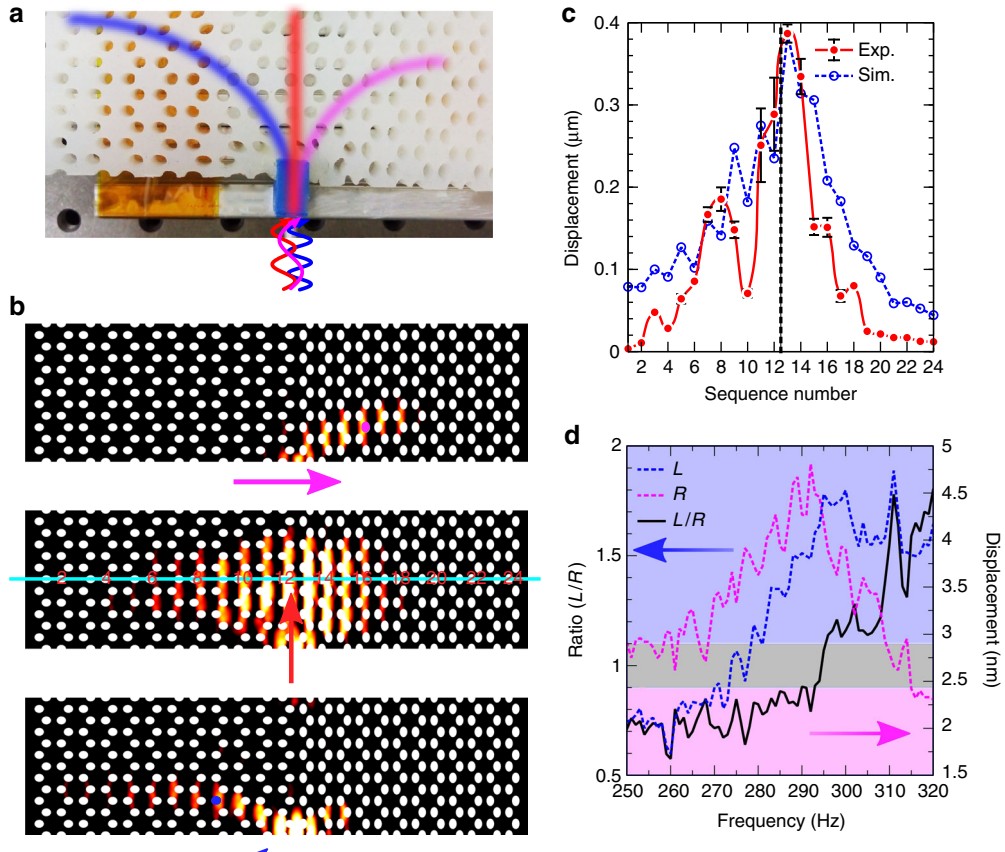

**Fig. 4** Experimental observation of topological interface state and demonstration of elastic wave splitter. **a** Experimental setup of a Ecoflex slab consisting with two elastic metamaterials, (0.6, 5.56%|0.68, −8.89%). Magenta, red, and blue lines schematically indicate elastic waves propagation directions as a function of input frequency. **b** Numerical simulations of vertical displacement fields with different transverse wave propagations at three input frequencies: 261, 274, and 286 Hz, from top to bottom. **c** Experimental observation of topological interface state at 274 Hz corresponding to the second panel in **b**, by measuring the displacement of the 24 holes marked by cyan line. Values of the measured displacements represent the mean of $n$ tests ($n = 5$, error bars are defined as s.d.). Simulation results are shown in blue dashed line. Sequence numbers marked in red indicate the hole numbers. The black dashed line indicates the position of interface between two metamaterials. **d** Experimentally measured displacement on the right side at the magenta hole is presented by the magenta dashed curve. Experimentally measured displacement on the left side at the blue hole is presented by the blue dashed curve. The displacement ratio of the left side over the right side, $L/R$ is presented as a function of input frequency (black solid curve). Magenta domain indicates the right propagation mode, while the blue domain reveals the left propagation mode. Gray region is the intermediate mode defined in main text

simulations and experiments in Supplementary Fig. 7 and Supplementary note 2 confirm the added mass effect can be neglected so that the estimated displacement field is valid.

To demonstrate the elastic wave splitter, the accelerometer is placed in the hole on the left or right side of the propagation pathway of the elastic wave according to the simulation results in Fig. 4b. The experimental setup is presented in Supplementary Fig. 9. Figure 4d presents the collected displacement of the left part (magenta dashed curve) and the right part (blue dashed curve) in the frequency range of 250 ~ 320 Hz. The calculated displacement ratio of the left part over the right part is illustrated in the black curve as shown in Fig. 4d. We define the area of the displacement ratio between 0.9 and 1.1 as intermediate mode (gray region in Fig. 4d), which corresponds to the most abrupt area in ratio curve in the frequency range of 293 ~ 295 Hz. The displacement with a ratio above 1.1 is regarded as left propagation mode, which is above 295 Hz. While the displacement with a ratio below 0.9 is regarded as right propagation mode, which is below 293 Hz. This feature may find application in phonon frequency splitter due to different group velocities as presented in the projected band structure (Supplementary Fig. 6), which differs from the chiral propagation in time-reversal breaking system[38].

The slight disagreement between the frequency of the intermediated mode (293 ~ 295 Hz) and that of the flat band (274 Hz) may be attributed to two reasons. First, the detection position where we put the accelerometer may affect the magnitude of the displacement. The vibration is mainly concentrated on the matrix rather than around the holes, so the measured displacement is slightly smaller than the simulation results. Second, the fixtures we put on the two edges of the interface line between two metamaterials may constrict the vibration at the vicinity of the clippers. In addition, a tiny intermediate region between two metamaterials arisen from the different strains setup may also affect the measurement of the displacement.

**Dynamical manipulation of topological interface states**. Tunable elastic topological state is important and may find application in large-scale phononic circuits. The zero-frequency adaptive behavior controlled by external mechanical loads has been displayed, with floppy modes and the states of self-stress[39]. Here we present a soft topological metamaterial with dynamically tunable topological properties. Figure 5a presents a combination of two elastic metamaterials, (0.6, $\varepsilon_1$|0.68, −8.89%), in which strain $\varepsilon_1$ is

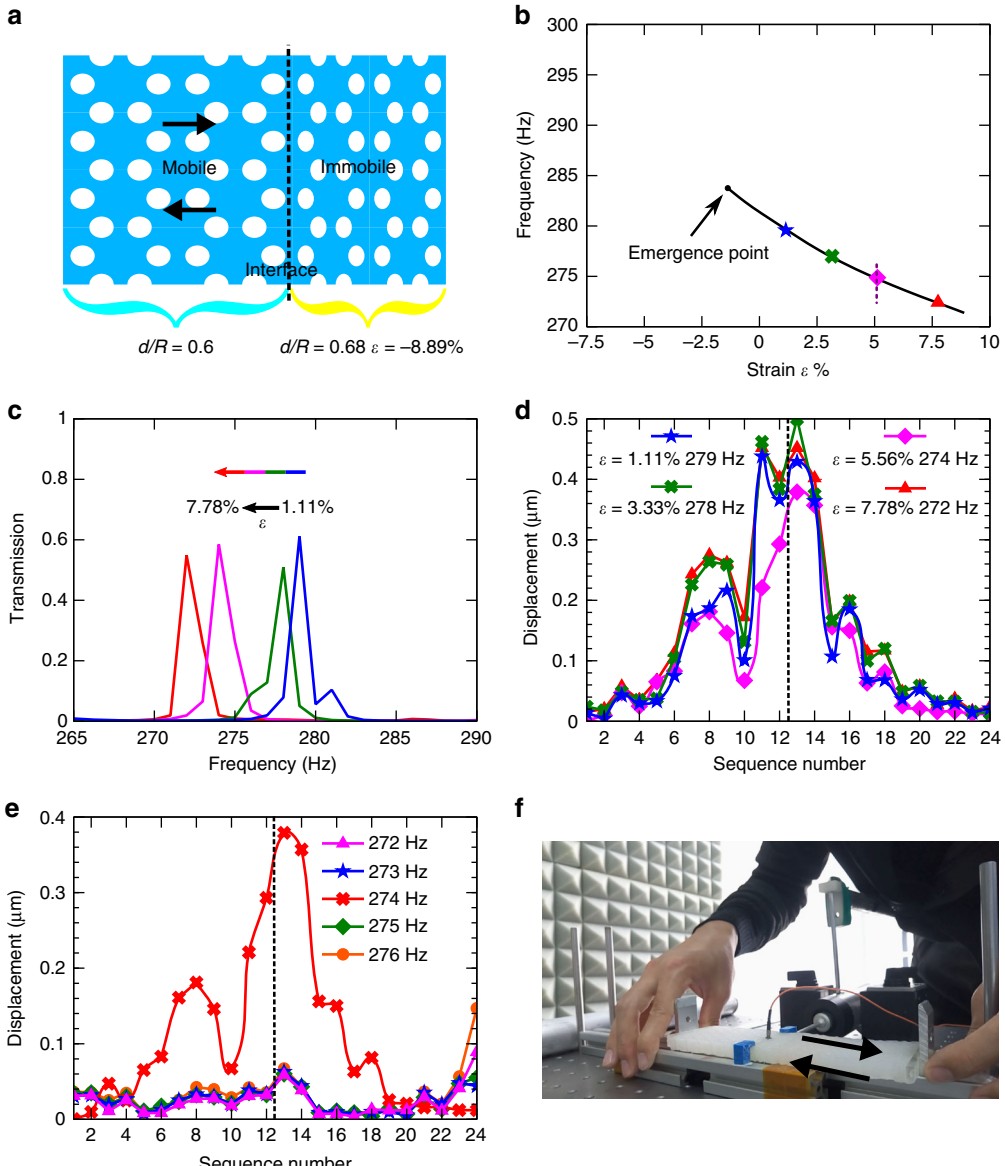

**Fig. 5** Tunability of soft topological system. **a** The schematic of tunable topological system with two metamaterials, (0.6, $\varepsilon_1$|0.68, −8.89%), where strain $\varepsilon_1$ is variable in the range of −5.56 ~ 8.89%. **b** The numerically simulated frequencies of topological interface state as a function of strain. The topological state emerges at a certain frequency and the frequency decreases as the strain increases. **c** The simulated transverse wave transmission peaks for four selected strains $\varepsilon_1 = 1.11, 3.33, 5.56$, and 7.78%, corresponding to the four colored symbols marked in **b**. **d** The experimentally measured vertical displacement field distributions at four selected strains. The markers and colors have their correspondences in **b**. **e** The experimentally measured displacement field distributions at five selected frequencies at strain of 5.56% along the purple dashed line in **b**, 272, 273, 274, 275, and 276 Hz. **f** A snapshot of experimental demonstration of dynamical manipulation of topological interface state. The black dashed lines in **a**, **d**, and **e** indicate the position of interface between two metamaterials

a variable in the range of −5.56 ~ 8.89%. According to the topological phase diagram (Fig. 2b), the metamaterial (0.68, −8.89%) at right side has a relatively large bandgap in the frequency range of 263 ~ 298 Hz. While the metamaterial (0.6, $\varepsilon_1$) at left side has a common gap with that at right side when $\varepsilon_1$ changes in the range of −1.11 ~ 8.89%. Figure 5b presents the numerically calculated topological interface states only emerged within the solid line in the frequency range of 271 ~ 285 Hz when strain $\varepsilon_1$ changes in the range of −1.11 ~ 8.89%. We selectively choose four topological interface states in the solid line (Fig. 5b) where strains $\varepsilon_1 = 1.11\%, 3.33\%, 5.56\%$, and 7.78%, corresponding to frequency of 279, 278, 274, and 272 Hz, respectively.

Figure 5c presents the simulated transmission spectra of transverse wave for the four selected strain levels, and all of them

have sharp transmission peaks. The transmission peak shifts to lower frequency or higher frequency when the metamaterial is stretched or compressed. Corresponding experimental displacement field distributions are displayed in Fig. 5d to confirm the existence of topological interface states using the experimental setup detailed in Supplementary Fig. 9. Besides, when we change the frequency of input excitation along the dotted line in Fig. 5b while keeping $\varepsilon_1$ fixed at 5.56%, the topological interface state can only be observed at the frequency of 274 Hz, as presented in the experimental displacement field distribution in Fig. 5e. The explicit comparison of experiment results and simulation results is displayed in Supplementary Fig. 8, where the good agreement of displacement field can be observed. The nature of elastomeric elastic system makes the deformation process continuously

reversible and repeatable, suggesting that elastic device can perform well after thousands of utilities.

The mechanical deformation induced appearance and disappearance of the strong field localization feature of interface states can be dynamically manipulated, as shown in Supplementary Movie. Figure 5f shows a snapshot of the movie, indicating the experimental setup and dynamical process. When the input vibration is fixed at frequency of 274 Hz, the topological interface states emerge at the strain of 5.56%; the interface states disappear when the strain is larger or smaller than 5.56%. The movie shows a time-dependent acceleration signal when we stretch or compress the metamaterial. The appearance of the topological interface states is observed when the acceleration value reaches the maximum, which is marked in the Supplementary Movie. The experiment details are shown in Supplementary note 2.

## Discussion

The transport of elastic wave in the elastic topological metamaterials is quite different from topologically protected transport in acoustic system. Although both elastic and acoustic waves can propagate along interface without backscattering, the working frequency of acoustic topological insulator is easily affected by its surroundings such as water and air[10]. Since the polarization of our elastic topological interface states is transverse, the working frequency is relatively stable. We further calculate the phase diagrams of soft metamaterial in water and vacuum background (Supplementary Fig. 10). We find that the band structure of the soft metamaterial in vacuum or water is similar to that in air, indicating a robust band structure regardless of chaotic fluid surroundings. We can find that the frequency ranges of bandgap of the soft metamaterials in air or vacuum are slightly different from that in water (~25 Hz), which mainly results from the impedance mismatch and different loss intensities at the boundary of soft metamaterial and its surroundings. Since the soft metamaterials in different surroundings share a similar phase diagram, we can easily find the topological interface states in a soft metamaterial even randomly patterned with different fillers (air, water, and vacuum) into holes, where the immune nature of topological states is reflected greatly. Our finding provides an example on topological insulator working in a complicated environment, which has special advantage in information processing and communication field.

Knowledge of strain-induced elastic topological phase transition opens avenues for topological state manipulation. This strategy gives us the possibility to realize and then control topological interface state statically and dynamically. Although in our study we only focus on unit cell of millimeter scale, the proposed design can be more complex and has various scales depending on the operating frequency. Our research may be generalized in other microscopic and macroscopic phononic systems such as thermal management and soft robotics that make better use of energy. Our study can also inspire electronic topological insulator used on elastomer substrate to develop flexible electronic devices[40].

## Methods

**Materials**. A commercial silicone rubber Ecoflex 0030 (Smooth on®) is used to cast the experimental samples with material density $\rho = 1030 \, \text{kg m}^{-3}$. In order to investigate the mechanical properties of Ecoflex 0030 (part A:part B = 1:1), a uniaxial tension experiment is carried out. A mold based on ASTM D412 standard is fabricated by 3 mm acrylic plate using a laser cutter, with a vector-optimized cutting path to avoid defect on the critical neck region. Subsequently, the Ecoflex mixture is cast into a two-part mold consisting of dog-bone-shaped top and acrylic plate bottom after thoroughly degassed in a vacuum chamber. Then, it is placed at room temperature for 4 h to be cured. Tensile test is performed on a universal testing machine and the relation between engineering stress and elongation is obtained (Supplementary Fig. 11). A nearly incompressible Yeoh hyperelastic

model is used to fit this relation, whose strain energy density is given by[41]:

$$W = \sum_{(i=1)}^{3} C_i (I_i - 3)^i + \frac{(J-1)^{2i}}{D_i} \tag{2}$$

Thus, we obtain the mechanical properties of the soft material as follows: $C_1 = 13.44 \, \text{kPa}$; $C_2 = 595 \, \text{Pa}$; $C_3 = -0.8153 \, \text{Pa}$; and $D_1 = D_2 = D_3 = 14.88 \, \text{GPa}^{-1}$.

**Sample fabrication**. In order to accurately fabricate the soft metamaterial sample, an assembled mold is prepared. The mold comprises a base, four lateral walls, and hundreds of well-polished stainless steel rods to shape the contour of holes precisely. The base and lateral wall are cut by a laser cutter from acrylic plate. Stainless steel rods with diameters of 3.0 and 3.4 mm are made by polishing and wire cutting to the height slightly higher than the walls on every side. Subsequently, these rods are inserted into the corresponding holes on the base. After the fabrication of mold, two parts of silicone rubber are mixed thoroughly by the ratio of 1:1 using an electric mixer and the casted mixture is placed in the vacuum chamber for degassing. Next, it is allowed to cure at room temperature for 4 h after pouring the mixture into the mold. According to this method, we can precisely fabricate sample with combination of elastic metamaterials with different filling ratios. After demolding, the sidewalls are cut from sample to satisfy periodic condition. Here we obtain a metamaterial sample with two 6 × 6 unit cells: $l = 10 \, \text{mm}$; $c = 5 \, \text{mm}$; $m = \sqrt{3}c$; and $d_1 = 3.4 \, \text{mm}$, $d_2 = 3.0 \, \text{mm}$ (Supplementary Fig. 1a).

**Testing and analysis**. The transverse wave is generated by a shaker (Brüel & Kjær, Type 5961) and vibration is transmitted by a rigid rod. At each side of sample, clips are used to fix, stretch, and compress sample. At the interface of two metamaterials, two clips are used to fix the edges of the interface to make the metamaterials move independently. We measure the interface state characterized by displacement field distribution using accelerometer (Brüel & Kjær, Type 4517) at several levels of applied deformation. At the strain level of interest, we immobilize the specimen by fixing the slide block on the slide guide and measure the displacement in a set of number-labeled air holes marked in Fig. 4b. In order to measure different propagation directions of elastic wave in two parts of metamaterials, we choose two holes on each side according to simulation results and put the accelerometer into them. The exciter provides a random signal and the spectra is obtained by Fourier transforms. The photo of specific experimental setup and related description are shown in Supplementary Fig. 9 and Supplementary note 2. The ambient noise is also recorded when the sample is statically placed.

**Numerical simulations**. Numerical simulations are performed by using COMSOL Multiphysics, a finite-element analysis and solver software. The simulations are implemented in the 2D acoustic-structure coupling module, including the actual geometric size and relevant properties. The system consists of the air resonator (vacuum and water) and soft material Ecoflex 0030. The parameters used for air are mass density 1.29 kg m⁻³ and sound speed 340 m s⁻¹. The mechanical properties for Ecoflex 0030 are extracted from Yeoh hyperelastic model fit with experiment data. The geometric parameters of unit cell are calculated as the elastic metamaterial is under small mechanical deformation (<9%) (Supplementary Fig. 1). The deformed unit cell with periodic boundary conditions is used to calculate the band structure and eigenmode, while the supercell with unidirectional periodic condition is used to calculate the projected band structure, detailed in Supplementary note 3. For calculating the transmission spectra, an external force with different frequencies is imposed on the boundary of finite size sample to mimic the incident wave. In addition, the displacement on the outgoing end is collected as the transmitted signal. The data line is set two to three lattice constants in front of the edge of perfect match layer (PML) depending on the frequency of input. For the calculation of wave propagation in finite sample, the PMLs are set around the sample to prevent the leakage of energy.

**Data availability**. The data in this study are available from the corresponding author on request.

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

## Acknowledgements

This work was supported by the National Natural Science Foundation of China No. 51572096, and the National 1000 Talents Program of China tenable in HUST. We thank Dr. Meng Xiao from Stanford University for fruitful discussions. We are grateful to Yugui Peng and Yaxi Shen from Huazhong University of Science and Technology for theoretical and experimental discussions.

## Author contributions

J.Z. and S.L. designed the project. S.L. performed the numerical simulation and carried out the experiments. D.Z. assisted with the numerical simulation. H.N. assisted with part of experiments. S.L. and J.Z. wrote the manuscrpt. All authors were involved in analysis and discussion of the results, and in improvement of the manuscript.

## Additional information

**Competing interests:** The authors declare no competing interests.

