## [Peer Review File · Nature Communications]

Reviewers' comments:

Reviewer #1 (Remarks to the Author):

The authors construct a soft elastic structure. This structure possesses gapped elastic bands with nontrivial topological numbers which can be altered by applying a nonlinear strain to the system. As the authors note in referencing one of Bertoldi's papers, closing a band gap by straining a system has previously been demonstrated. Similarly, they note that the topological state without tuning has previously been realized. Additionally, Chen Upadhyaya Vitelli 2014 and Rocklin Zhou Sun Mao 2017 have used nonlinear motions to tune between topologically distinct zero-frequency states. As such, the present work represents a natural but somewhat incremental synthesis of recent work.

The authors mention a number of potential applications: elastic energy storage, elastic wave guide technology, earthquake "prevention" (protection?), large-scale phononic circuits, thermal management, electronic topological insulators, soft robotics and broadly, military, communications and information processing applications. It is not clear from the present manuscript how the experimentally demonstrated device would apply in these areas. The simulation appears to show that the device has a tunable ability to filter out waves at certain frequencies. How strong is this filtering, and does it compare favorably to other, non-topological ways of filtering out modes of certain frequencies?

Symbol introduced on line 115: Given its centrality, the meaning and significance of this symbol needs to be explained in the main text and/or the figure captions, rather than requiring consultation of reference 33. Both that reference and the supplementary material seem to define the symbol as applying to a particular band (denoted by (n)), while the text seems to refer to it as associated with the band gap and does not give it an index.

The authors also need to explain their results more clearly in the figures. In particular:

1C,D: What is the source of this data? Simulation?

2A: Do the band gap signs apply to the blue and yellow regions? They appear to indicate the white regions.

2B: What is this diagram saying? It appears to indicate that the topological state (yellow or blue) depends on frequency but not filling fraction. I would think that the topological state would be defined for a given filling fraction and strain.

What do the purple and orange stars mean? Are these stars in 2B the ones that are referenced in the caption as being in 2A?

3B: What do the overbars over Gamma and M denote in the figure?

3C: It would be helpful, though not necessary, for the interface and bulk states to be marked here as in 3B.

4B: The "sequence number" marked in light blue is not legible. It's important that it be legible, particularly because sequence number, as opposed to cell number, is a somewhat unusual index.

Do the authors ever experimentally measure transmission? Figure 4D appears to be experimental data from a source at the interface, and figures 3C and 5C appear to be simulation data. If the transmission can't be done experimentally, why can't it? Do the authors expect that their device would have the transmission properties predicted by simulation?

The authors provide clear descriptions of their experimental parameters and methods that appear to be sufficient to reproduce the work.

The references appear to be largely on point, though reference 21 seems to be referenced for "earthquake prevention", and doesn't have an obvious relationship to earthquakes.

Reviewer #2 (Remarks to the Author):

There are many commendable aspects of this work: there is considerable interest in mechanical topological materials, and the demonstration of a tunable edge states is quite novel. Some version of this work deserves to be published in a high profile journal.

Unfortunately, I have two main concerns about the manuscript in it's current form:

1) Clarity. The paper suffers from numerous grammatical errors and strange word choices. This severely impacts the readability of the manuscript. Apart from this, the paper is not laid out in a clear and logical manner, and in several places it is unclear if they are talking about experiment or simulations! I'm not sure what the supplementary movie is supposed to show. I would encourage the authors to rewrite the paper with a logical structure, and pay more attention to the clarity of their writing.

2) Quality of the experimental results, about which I have two issues:

2a) You can not have a material compressed on one side and stretched on the other without clamping along the edge, which would certainly be expected to modify the results! Indeed, it is clear in the images (Fig 4a) that there is an intermediate region in between, rather than an abrupt change in strain (as in the simulations), but there is no discussion of the effect of this important difference. Should we even still expect an edge state here? The data indicates there is some confined mode, but I see no evidence that it is topological in nature. Moreover, the data for frequency splitting (Fig 4d) is very weak: the variation in both directions with frequency is much stronger than the difference between them, while the splitting seems nearly perfect in the modelling (Fig 4b).

2b) There is no discussion of the effect of placing an accelerometer in the lattice. Without more information, I would expect the added mass to totally change the results, as presumably the accelerometer density is comparable to that of the material system. In this case, each data point is essentially measuring a different system, and so the results can not be interpreted as the authors would like. This needs to be addressed directly, either with some order of magnitude estimate as to why this should not be a problem, or by using a better method to measure vibration (either a high speed camera or laser vibrometer come to mind).

Finally, I would like to again emphasize that I think this is very interesting work, but it is flawed in it's current form. If the authors could address these issues, I would be happy to see this paper published.

Reviewer #3

Report: NCOMMS-17-15705, of Li *et al.* 'Observation of elastic topological states in soft materials'.

The authors report a new class of two dimensional soft mechanical metamaterials made of holes positioned in a honey-comb lattice. This metamaterial presents topological states that can be switched on and off and also more finely tuned by either changing the hole size or by applying and external static compression/traction. They investigate numerically and experimentally the propagation of elastic waves in this architected medium. More particularly the authors design a hybrid beam composed of two metamaterials. The topological state corresponds to a localization of the mechanical energy at the interface. They also show that by exciting the system close to the interface, the wave can be sent right side or left side, that is called beam splitter by the author.

The authors perform an impressive experimental and numerical realization and the results represent an important contribution for the burgeoning field of mechanical metamaterials. The beam splitter of elastic waves is a striking and very smart realization. However, I have major concerns regarding the origin of these topological states. The author does not even try to discuss it, which is a major problem of the manuscript. An important effort has to be made on this point. Moreover, the discussion of the figures is extremely hard to follow and disappointing. I may recommend publication in Nature communications providing the authors deeply revise the presentation of the manuscript.

The authors will find below a series of major and minor concerns

Major concerns

Where do the topological properties come from? To make my question clear, I do not deny that topological properties arise, this is clearly demonstrated by the measure of the Zak phase. However, the author should be able to answer what is the physical origin of this topology? What is the mechanism. I think it has to do with particular geometrical conditions in the unit cell at the critical filling ratio/load ratio. A schematic of principle should come along with Fig1b or Fig2c. This point is by far my major criticism.

The figures are extremely hard to understand because the relevant information is awkwardly split between the text and the caption. To exemplify my remark, let us take Fig5

- Fig 5 b. What are the black, red curves? Are they from simulations? What are the different symbols? It should be in the captions.
- Fig 5c What are the different curves?
- Fig 5d and e. What means "sequence"? Where are the displacements measured? Are we talking about longitudinal or lateral displacements? Where is the interface? The author

could draw a horizontal line in Fig 5a to show where it is measured. The author should also draw a vertical line in Fig 5d and e showing the interface position. Fig 5f should be an inset of Fig5a?

- Fig5 b and 5c should be exchanged.
- After three careful readings, I do not still even understand what Fig5 is supposed to demonstrate.

I do understand the difficulty of realizing the beam splitter in Fig 4. The efficiency is close to 50 %. Do the authors have an idea if this is due to a limitation of the material (dissipation?), or a finite size effect etc..? Or is there any fundamental limit?

The author should discuss (or at least mention) the relevance of their work in the framework of very recent experimental realization. Especially

- For the beam splitter part:

Topological sound in active-liquid metamaterials, Souslov *et al*, *Nature Physics* (2017)

Is there any relation between what these authors called circulation and your system which could explain the direction of propagation of sound

- For the design:

Programmable Mechanical Metamaterials. Florijn *et al.*, *Phys. Rev. Lett.* 113, 175503 (2014).

- For the topological interface

Selective buckling via states of self-stress in topological metamaterials, Paulose *et al.*, *Proceedings of the National Academy of Sciences* (2015)

Minor concerns

All the caption should be expanded. The colors, symbol, if it experimental or numerical (etc...) should appear in the caption.

Abstract, line 18. "arbitrary", please rephrase.

Page 2 line 45 "Elastic waves,....., have "

Page 2 line 48 "which is ... prevention" It is overstated no? It could be useful, but at the current stage, it is far for real earthquake prevention.

Page 2 line 52 "advanced fabrication technique", please give some concrete example.

Page 5 line 144 "Put together', please rephrase

Page 7 line 192-194. Please rephrase. Why is there a criterion 0.9-1.1. It seems a bit arbitrary.

Page 7 line 206-208 Please rephrase. After three careful reading, I still do not understand what the authors means.

Part “dynamical manipulation of topological interface”. I do not know why but this part is poorly written and not well organized. There is back and forth between figures (5a 5b 5c 5d 5b 5e 5f 4a 5b 4b) which make it extremely hard to understand. After careful readings, I still do not understand what Fig5 is supposed to demonstrate.

Page 7 line 208 “Obviously” Please remove. Nothing is a priori obvious.

Page 8 line 216 “at four strain...” it should “for four”.

Page 8 231-232 “Actually.... ” What mean the author mean? Please rephrase

Page 8 line 235 “transport along the interface”. Please rephrase as one could understand you are talking about edge states (which is not the case here).

Page 9 line 268 “potential in military and communication field”. Please give some examples, otherwise this sentence is purposeless. I would not have brought into light military applications, but on this particular point I must admit that it is only my personal opinion and taste.

Movie What it is the purpose of the movie? It shows a sort of fluctuation of a signal on oscilloscope which measured an acceleration. It seems that there is an upper threshold, but I have no idea what it should refer to. The author should make an effort of presentation

Supplementary Fig 2b (upper colorplot). The colorplot does not present a top/down symmetry. Why is this?

Reviewer #4 (Remarks to the Author):

In this paper, authors present an experimental observation with numerical simulation of tuneable topological states in soft elastic metamaterials. They show that they can switch the topological phase by changing filling ratio, tension or compression of the elastic metamaterials. To convince readers they provide a topological phase diagram of elastic metamaterials under deformation. Even if the paper looks well written in the introduction, I think it is difficult to deeply understand scientific arguments of the authors. Very often one has the impression that it is somehow as the authors are saying "can be formed accidentally". I would avoid such statement, as the content of the paper is indeed very technical and not so easy to obtain. It give the impression that the main strategy, if it exist, is "fulling around". I think the paper should be improved significantly and then should be publishable even in Nature Commun ...

Now technically, there is a mystery to me, how they connect the hyperelastic model to their structures. It is not enough to know the elastic law. Under a constrain, not all contributions in a microstructure will be "deformed" by the same way. I would recommend to the authors to read the work of William Parnel (Univ of Manchester) who deeply studied this effect.

For the numerical model, it is not clear to me what they have really done. I have worked with Comsol for 10 years and such study is far from been trivial. I would appreciate some more details in this part. How they deal with overall constrain and with the post linearization of the problem. In figure 1, what is longitude in "Dash lines indicate longitude wave bands". Please explain how you get or define the Zak phase.

Figure 2 is nicely colourful but what is the meaning of each colour? Could you add a legend to the colours?

Figure 3, please specify the source term and its real position. Generally it is a bad idea to have a source or recording term on the edge with a PML. It always depends on the definition of the PML, which in mechanics can be less efficient than in electromagnetism. What is the bar on top of Γ and M in the panel b?

Figure 4, be more precise in the caption. I have no clue how this was done nor what is really shown by reading the caption.

Figure 5, I do not understand the panel (a). What is the physical origin of the "interface"? what are the picks in the panel c? that looks as single point picks which is not acceptable as measurement from the resolution point of view.

Reviewer #1

The authors construct a soft elastic structure. This structure possesses gapped elastic bands with nontrivial topological numbers which can be altered by applying a nonlinear strain to the system. As the authors note in referencing one of Bertoldi's papers, closing a band gap by straining a system has previously been demonstrated. Similarly, they note that the topological state without tuning has previously been realized. Additionally, Chen Upadhyaya Vitelli 2014 and Rocklin Zhou Sun Mao 2017 have used nonlinear motions to tune between topologically distinct zero-frequency states. As such, the present work represents a natural but somewhat incremental synthesis of recent work.

We thank the reviewer's positive comments to our results and providing substantial literature background on our work. As the reviewer mentioned, our works focus on the experimental and numerical realization of tunable elastic topological states by mechanical deformation in elastic metamaterials. Following the reviewer's suggestion, we have carefully revised the manuscript.

The authors mention a number of potential applications: elastic energy storage, elastic wave guide technology, earthquake "prevention" (protection?), large-scale phononic circuits, thermal management, electronic topological insulators, soft robotics and broadly, military, communications and information processing applications. It is not clear from the present manuscript how the experimentally demonstrated device would apply in these areas.

We thank the reviewer's comments on the feasibility of putting elastic waves and/or topological effects to work in actual applications.

First of all, we mentioned in Introduction session in the manuscript that elastic waves have three potential applications (elastic energy storage, elastic wave guide technology, earthquake protection), which were supported directly or indirectly by references (*Sensors-Basel* **13** 8352-8376 (2013), *Phys. Rev. Lett.* **112** 133901 (2014), *Adv. Mater.* **28** 5943-5948 (2016)).

Secondly, the elastic topological devices possess the back-scattering-free feature, which is useful to realize large-scale phononic circuits. While the existing large-scale phononic circuits using conventional waveguides are limited by scattering due to defects or bent.

Thirdly, if the frequency of elastic waves is high enough to reach phonon frequency scale, the interface acts essentially as a heat transport channel, suggesting a potential application in thermal management.

As for the electronic topological insulators, there have been several literatures about strain-induced topological transition (*Phys. Rev. B* **94**, 085417 (2016), *Sci. Rep.* **5**, 17980 (2015)).

Soft robotics is an emerged and important application of soft materials. Progress in soft materials will potentially impact on the development of soft robots. The topological zero mode in soft materials has controlled mechanical states and/or floppy modes, which is potentially useful in the application of soft robotics.

We revised our manuscript to make it clear that the potential applications arisen from taking advantage of elastic waves and/or topological states in soft materials. We also deleted the sentence involving the potential applications in military, communications, and

information processing applications, to make it more appropriate in stating the potential applications.

The simulation appears to show that the device has a tunable ability to filter out waves at certain frequencies. How strong is this filtering, and does it compare favorably to other, non-topological ways of filtering out modes of certain frequencies?

We thank the reviewer's comments. Our work presents the tunable capability to filter out elastic waves at certain frequencies, as depicted by the transmission peaks in Fig. 5c. Transmission peaks in topological band are a significant characteristic of topological devices. The calculated transmissions of the four filters in our devices are all over 50%. We do agree that the filtering capability comparison between topological and non-topological ways is an interesting topic. However, non-topological filters usually involve various mechanisms, such as defect mode and local resonance flat band. Thus it will be very hard to conduct a simple comparison. Moreover, the in-depth study on the capability of filters is a rich enough topic to warrant its own paper.

Symbol introduced on line 115: Given its centrality, the meaning and significance of this symbol needs to be explained in the main text and/or the figure captions, rather than requiring consultation of reference 33. Both that reference and the supplementary material seem to define the symbol as applying to a particular band (denoted by (n)), while the text seems to refer to it as associated with the band gap and does not give it an index.

We thank the reviewer's kind suggestion. Following the reviewer's suggestion, we explained the symbol in the main text and the figure caption of Fig.1. In addition, we added the description of the symbol ς in the text as the band gap sign to characterize the topological property of band gap. We keep the discussion of symbol ς and the calculation method in the Supplementary Material.

The revised manuscript in page 4 paragraph 2 is attached as below:

“Given the mirror symmetry of our physical system, the Zak phase is quantized and ensured to characterize the topology of the bulk. For each bulk band, the Zak phase should be π if the eigenmode at center of Brillouin zone possesses different symmetries with that at edges. Otherwise, the Zak phase is 0.”

The revised manuscript in page 5 paragraph 1 is attached as below:

“We can similarly obtain the $\text{sgn}(\varsigma)$ of band gap associated with Zak phase by a simple expression³⁴:

$$\text{sgn}(\varsigma^{(n)}) = (-1)^n (-1)^l e^{i \sum_{m=0}^{n-1} \theta_m^{\text{Zak}}}$$

and where n is the sequence of band gap and l is the number of crossing points beneath this band gap. The resultant band gap signs are marked in Fig. 1c and Fig. 1d. The analysis details are seen in Supplementary note 1.”

The revised figure caption of Fig. 1 is attached as below:

“Figure 1 | Design of topological elastic metamaterials and two band inversion processes. a, Soft elastic metamaterial with periodic honeycomb holes of air in a rectangle silicon rubber (Ecoflex). The inset shows the stretchability of the soft metamaterial. **b,** The schematic of our soft elastic metamaterials. Red dashed hexagon is

*the primitive cell with the hexagon edge length R and hole's diameter d . \vec{a}_1 and \vec{a}_2 are the basic vectors. **c**, Band inversion process as a function of filling ratio of d/R without applied strain. The filling ratios of d/R chosen are 0.6, 0.7156 and 0.78. Dashed lines indicate longitudinal wave bands. Solid lines and dotted lines indicate transverse wave bands in ΓM direction and other directions, respectively. Inset is the irreducible Brillouin zone. **d**, Band inversion process as a function of strain with a fixed filling ratio of $d/R = 0.68$. Three strains chosen from compression to tension are -4.44% , -1.53% and 3.16% . The calculated Zak phase could be 0 and π , which is marked on the corresponding bulk band in ΓM direction (solid lines) in **c** and **d**. The calculated band gap signs ζ are marked in the corresponding band gap. All band structures are from numerical simulation."*

The authors also need to explain their results more clearly in the figures. In particular:

We thank the reviewer's suggestion on figure captions. Following the reviewer's suggestion, we carefully revised the figure captions.

1C,D: What is the source of this data? Simulation?

We thank the reviewer's comment. The band structures shown in Fig. 1c and Fig. 1d are from numerical simulation. We revised the figure caption of Fig. 1 by adding a sentence "All band structures are from numerical simulation".

2A: Do the band gap signs apply to the blue and yellow regions? They appear to indicate the white regions.

We thank the reviewer's comment. Indeed, Fig. 2a can be misunderstood. We added arrows in Fig. 2a to clarify that the band gap signs indicate the blue and yellow regions respectively. In addition, we revised the corresponding figure caption.

The revised figure caption is attached as below:

"a, The frequencies of two states at M point as a function of filling ratio of d/R without applied strain. The band gap signs of the cyan and yellow regions are $\zeta > 0$ and $\zeta < 0$, respectively."

2B: What is this diagram saying? It appears to indicate that the topological state (yellow or blue) depends on frequency but not filling fraction. I would think that the topological state would be defined for a given filling fraction and strain.

We thank the reviewer's comments on the topological phase diagram. For the time reversal preserving topological insulators, not only are the topological phases of two parts different, but also do frequencies of band gaps overlap. Actually, the topological phases of two parts in our system are controlled by two factors: strain and filling ratio. Thus, there are three different factors involved to regulate the topological states in our system: strain, filling ratio and frequency. All of the three factors are included in the topological phase diagram, as shown in Fig. 2b.

What do the purple and orange stars mean? Are these stars in 2B the ones that are referenced in the caption as being in 2A?

We thank the reviewer's comments. These stars correspond to the band-edge states we choose to show the vibration modes in Fig. 2c. Specifically, two stars on the left side represent the state $\varepsilon = -8.89\%$, $d/R = 0.68$ and two stars on the right side represent the state $\varepsilon = 5.56\%$, $d/R = 0.6$. The vibration modes are shown in Fig. 2c and the frame colors correspond to star colors. From the inversion of vibration mode, we can explicitly observe topological phase transition. We revised the typo in the figure caption. These stars are referenced in the figure caption of Fig. 2c.

The revised figure caption of Fig. 2 is attached as below:

“Figure 2 | Topological phase diagram. a, The frequencies of two states at M point as a function of filling ratio of d/R without applied strain. The band gap signs of the cyan and yellow regions are $\zeta > 0$ and $\zeta < 0$, respectively. **b**, Topological phase diagram as a general design scheme consists of two domains separated by topological transition line (black dash-dotted line) that is formed by connecting the topological transition points of different filling ratio systems. Green, blue and red solid lines are obtained by investigating the frequencies of two band-edge states at M point as a function of strain. The band gap signs ζ are marked in the yellow and cyan regions. **c**, Real parts of eigen vertical vibration modes at M point inversion represents topological phase inversion in solid. Left panel shows the vibration modes under strain $\varepsilon = -8.89\%$ and filling ratio of $d/R = 0.68$. Right panel shows them at $\varepsilon = 5.56\%$, $d/R = 0.6$, marked by stars in **b**. Corresponding stars are beside the vibration modes. The imaginary part of surface impedance and band gap sign are marked between two vibration modes. All data are from numerical simulation.”

3B: What do the overbars over Gamma and M denote in the figure?

We understand the concern of the reviewer. It is the common practice in calculation of projected band structure to write a bar above Γ and M, which distinguishes the Γ and M from band structure of unit cell (*Phys. Rev. B* **90**, 075423 (2014)).

We revised the figure caption of Fig. 3 by rewriting the sentence “The simulated projected band structure along k_x direction indicated by $\bar{\Gamma}\bar{M}$ with transverse interface modes. The bars above Γ and M are used to distinguish the Γ and M from band structure of unit cell.”

3C: It would be helpful, though not necessary, for the interface and bulk states to be marked here as in 3B.

We thank the reviewer's good suggestion. We colored the bulk states with magenta corresponding to the projected band structure in Fig. 3b. We also marked the interface and bulk states in Fig. 3b to make it clear.

4B: The "sequence number" marked in light blue is not legible. It's important that it be legible, particularly because sequence number, as opposed to cell number, is a somewhat unusual index.

We thank the reviewer's good suggestion. We changed the color of the “sequence number” to red and enlarged the font size to make them legible.

Do the authors ever experimentally measure transmission? Figure 4D appears to be experimental data from a source at the interface, and figures 3C and 5C appear to be simulation data. If the transmission can't be done experimentally, why can't it? Do the authors expect that their device would have the transmission properties predicted by simulation?

We thank the reviewer's comments. Fig. 4d is from the experimental measurement and specific method is written in Method section. Fig. 3c and Fig. 5c are from numerical simulation. In fact, we cannot measure the transmission spectrum experimentally as our simulations predict because the claps used to stretch and compress the sample are fixed on both sides of sample. It needs a smarter gripping approach to experimentally realize the transmission properties predicted by simulation.

The authors provide clear descriptions of their experimental parameters and methods that appear to be sufficient to reproduce the work.

We thank the reviewer's positive comments.

The references appear to be largely on point, though reference 21 seems to be referenced for "earthquake prevention", and doesn't have an obvious relationship to earthquakes.

We thank the reviewer's comments. We rewrote the sentence in page 2 paragraph 2: *"Through creating band gaps in architected materials with periodic porous structures, elastic wave can be dramatically attenuated, which is particularly useful in vibration isolation²¹".*

Reviewer #2

There are many commendable aspects of this work: there is considerable interest in mechanical topological materials, and the demonstration of a tunable edge states is quite novel. Some version of this work deserves to be published in a high profile journal. Unfortunately, I have two main concerns about the manuscript in it's current form:

We thank the reviewer's positive comments to our results and novelty. Following the reviewer's suggestion, we have carefully revised the manuscript.

1) Clarity. The paper suffers from numerous grammatical errors and strange word choices. This severely impacts the readability of the manuscript. Apart from this, the paper is not laid out in a clear and logical manner, and in several places it is unclear if they are talking about experiment or simulations! I'm not sure what the supplementary movie is supposed to show. I would encourage the authors to rewrite the paper with a logical structure, and pay more attention to the clarity of their writing.

We thank the reviewer's comments on clarity of our manuscript. Following the reviewer's suggestion, we revised all the grammatical errors and replaced the strange words. We revised all the figure captions to explicitly show the results from experiments

or simulations. We also revised the session of dynamical manipulation of topological interface state and made it more readable and logical. For the Supplementary Movie, we added detailed explanations and marks to make it clear.

2) Quality of the experimental results, about which I have two issues:

2a) You can not have a material compressed on one side and stretched on the other without clamping along the edge, which would certainly be expected to modify the results! Indeed, it is clear in the images (Fig 4a) that there is an intermediate region in between, rather than an abrupt change in strain (as in the simulations), but there is no discussion of the effect of this important difference. Should we even still expect an edge state here?

We thank the reviewer's comments. In our experimental setup, we use two clips to fix the two edges of the interface of two metamaterials rather than fixing the interface, to minimize the side effect. We revised our Method session, Supplementary note 2 and the caption of Supplementary Fig. 7 to make it clear.

The revised Method session is attached as below:

"At the interface of two metamaterials, two clips are used to fix the edges of the interface to make the metamaterials move independently."

The revised caption of Supplementary note 2 is attached as below:

"...and the other is glued at two edges of the interface to realize independent motion of each soft metamaterial (blue 3D printing PLA in Supplementary Fig. 7)."

The revised caption of Supplementary Fig. 7 is attached as below:

"In addition, fixtures are glued to two edges of the interface and the slide guide using cyanoacrylate adhesive."

The numerical simulation model is formed by two metamaterials with different topological phases, while there exists an intermediate region in current experiment setup when the metamaterial is under tension or compression. As long as the two parts own different topological phases, the topological state is expected to appear at the interface. The intermediate region can be considered as a metamaterial with the continuously tuned topological phase, whose gradient is very large and area is small. Therefore, according to the perfect match between simulation and experiment, we believe that the topological states still exist. We also revised our manuscript by discussing the intermediate region.

The revised manuscript in page 8 paragraph 1 is attached as below:

"Second, the fixtures we put on the two edges of the interface line between two metamaterials may constrict the vibration at the vicinity of the clippers. In addition, a tiny intermediate region between two metamaterials arisen from the different strains setup may also affect measurement of the displacement."

The data indicates there is some confined mode, but I see no evidence that it is topological in nature.

We understand the reviewer's concern. From the eigen mode analysis, two metamaterials have distinct topological phases. Further, we calculate the Zak phase of the first and the

second bulk band along ΓM direction, which confirms the band inversion. According to the simulation results in Fig. 3b and Supplementary Fig. 6, there is a topological interface mode at the frequency of 274 Hz. Therefore, this localized mode indicated by the experimental data is the topological interface mode.

Moreover, the data for frequency splitting (Fig 4d) is very weak: the variation in both directions with frequency is much stronger than the difference between them, while the splitting seems nearly perfect in the modeling (Fig 4b).

We thank the reviewer's comment on beam splitter. It is difficult to realize the beam splitter in elastic system. The difference between simulation and experiment may attribute to two reasons. First, during the experimental measurement, we put the accelerometer in the hole rather than in the matrix. While the simulated displacement field is mainly concentrated on matrix instead on holes. Second, the fixtures we put on the two edges of the interface of the elastomer sample form a small region of intermediated area, which may affect the experimentally measured displacement. Therefore, the experiment result is less obvious than the simulation result due to the limitation of our experiment equipment. We revised the manuscript by adding the explanations.

The revised manuscript in page 8 paragraph 1 is attached as below:

“First, the detection position where we put the accelerometer may affect the magnitude of the displacement. The vibration is mainly concentrated on the matrix rather than around the holes, so the measured displacement is slightly smaller than the simulation results. Second, the fixtures we put on the two edges of the interface line between two metamaterials may constrict the vibration at the vicinity of the clippers. In addition, a tiny intermediate region between two metamaterials arisen from the different strains setup may also affect the measurement of the displacement.”

2b) There is no discussion of the effect of placing an accelerometer in the lattice. Without more information, I would expect the added mass to totally change the results, as presumably the accelerometer density is comparable to that of the material system. In this case, each data point is essentially measuring a different system, and so the results can not be interpreted as the authors would like. This needs to be addressed directly, either with some order of magnitude estimate as to why this should not be a problem, or by using a better method to measure vibration (either a high speed camera or laser vibrometer come to mind).

We understand the reviewer's concern about the quality of our experimental results using accelerometer. The added mass will absolutely affect the experimental results. But the effect is tolerable because the effective density of accelerometer (0.0084 g/mm^3) is very small compared to the density of the sample (1.03 g/mm^3), which is only about 0.82%. We acknowledge that contactless methods will be helpful, such as high-speed camera or laser vibrometer, but limited by the experiment facilities, it is very difficult for us to conduct such experiment. Actually, our experimental results agree well with the simulation results, indicating the validity of our current experimental setup. We revised our manuscript to include the discussion of this effect in page 7 paragraph 2.

The revised manuscript is attached as below:

“Note that inserting an accelerometer into the hole to measure the displacement will bring added mass to the sample. Considering the relatively low effective density of accelerometer (0.0084 g/mm^3) compared with the density of the elastomer sample (1.03 g/mm^3), this effect is thus negligible.”

Finally, I would like to again emphasize that I think this is very interesting work, but it is flawed in its current form. If the authors could address these issues, I would be happy to see this paper published.

We thank the reviewer's admiration to our work. We acknowledge there are some flaws in our manuscript. Following the reviewer's useful suggestions, we have revised the manuscript and addressed the questions thoroughly.

Reviewer #3

The authors report a new class of two dimensional soft mechanical metamaterials made of holes positioned in a honey-comb lattice. This metamaterials presents topological states that can be switched on and off and also more finely tuned by either changing the hole size or by applying an external static compression/traction. They investigate numerically and experimentally the propagation of elastic waves in this architecture medium. More particularly the authors design a hybrid beam composed of two metamaterials. The topological states corresponds to a localized of the mechanical energy at the interface. They also show that by exciting the system close to the interface, the wave can be send right side or left side, this is called beam splitter by the author. The authors perform an impressive experimental and numerical realization and the results represent an important contribution for the burgeoning field of mechanical metamaterials. The beam splitter of elastic waves is a striking and very smart realization.

However, I have major regarding the origin of these topological state. the author does not even try to discuss it, which is a major problem of the manuscript. An important effort has to be made on this point. Moreover, the discussion of the figures is extremely hard to follow and disappointing. I may recommend publication in Nature communications providing the authors deeply revise the presentation of the manuscript.

We thank the reviewer's positive comments on our research. We are also glad that the reviewer summarizes several substantial points of our work. In the following, we will address the reviewers' concerns point by point and revise our manuscript.

The authors will find below a series of major and minor concerns.

Major concerns

Where the topological properties come from? To make my question clear, I do not deny that topological properties arise, this is clearly demonstrated by the measure of Zak phase. However, the author should be able to answer what is the physical origin of this topology? What is the mechanism. I think it has to do with a particular geometrical conditions in the unit cell at the critical filling ratio/load ratio. A schematics of principle should come along with Fig1b or Fig2c. This point is by far my major criticism.

We thank the reviewer's comment. In the main text and Supplementary materials, we have shown the numerical calculation to evaluate the Zak phase of certain reduced 1D bands for a particular $k_{||}$ by symmetry analysis. It is the mirror symmetry of our system that ensures and quantizes the Zak phase (0 or π) to characterize the topology of bulk. For a one-dimensional periodic system with inherent mirror symmetry, the value of the geometric Zak phase in a bulk band is related to the sign of reflection phase inside the band gap, which corresponds to the sign of band gap (*Phys. Rev. B* **90**, 075423 (2014)). Further, the reflection phase is associated with surface impedance $Z(\omega, k_{||})$. Actually, Fig. 2b can be considered as a surface impedance diagram as a function of strain. One region with $\zeta < 0$ has $Z(\omega, k_{||}) < 0$ and the other region with $\zeta > 0$ has $Z(\omega, k_{||}) > 0$. This property, together with interface state formation condition of $Z_L(\omega, k_{||}) + Z_R(\omega, k_{||}) = 0$, implies that there must exist an interface state inside the common gap if the surface impedances of the two metamaterials have different signs (*Phys. Rev. B* **90**, 075423 (2014)). Whereas, the distinct topological polarizations have been reported in zero frequency states, corresponding to different lattice geometries, such as bond length. (*Nat. Phys.* **10**, 39–45 (2014), *Nat. Comm.* **8**, 14201 (2017)). We revised the main text by adding the origin of the bulk topology and formation of topological interface state. In order to schematically show the principle of the topological interface state formation condition, we added the surface impedance marks to Fig. 2c.

The revised manuscript in page 4 paragraph 2 is attached as below:

“Given the mirror symmetry of our physical system, the Zak phase is quantized and ensured to characterize the topology of the bulk.”

The revised manuscript in page 5 paragraph 2 is attached as below:

“Since the band gap sign is related to the reflection phase and further associated to the surface impedance $Z(\omega, k_{||})$, the topological phase diagram can be considered as the surface impedance diagram as a function of strain. One region (yellow region) with $\zeta < 0$ has $Z(\omega, k_{||}) < 0$ and the other region (cyan region) with $\zeta > 0$ has $Z(\omega, k_{||}) > 0$.”

The revised manuscript in page 6 paragraph 2 is attached as below:

“When two elastic systems with different topological invariants are edge-to-edge joint, the topological interface states can be predicted to emerge according to the surface impedance match condition $Z_L(\omega, k_{||}) + Z_R(\omega, k_{||}) = 0$.”

The revised figure caption of Fig. 2 is attached as below:

“The imaginary part of surface impedance and band gap sign are marked between two vibration modes.”

The figures are extremely hard to understand because the relevant information is awkwardly split between the text and the caption. To exemplify my remark, let us take Fig5

We thank the reviewer's comment on clarity of main text and figure captions. We revised the text and figure captions to make them clear, not limited in Fig. 5 and paragraph of dynamical manipulation of topological interface state.

Fig 5b. What are the black, red curves? Are they from simulations? What are the different symbols? It should be in the captions.

We thank the reviewer's comment. The black curve represents the evaluation of frequency of the upper band gap edge and the red curve represents that of the lower band gap edge. Given its being misunderstood, we delete the dashed line and only keep the black solid line that indicates the frequency of interface state as a function of strain. We also revised the related part in main text. The data in Fig. 5b are from simulation and the different symbols are referred to Fig. 5c and Fig. 5d.

The revised manuscript in page 8 paragraph 2 is attached as below:

“Fig. 5b presents the numerically calculated topological interface states only emerged within the solid line in the frequency range of 271 ~ 285 Hz when strain ε_1 changes in the range of -1.11% ~8.89%. We selectively choose four topological interface states in the solid line (Fig. 5b) where strains $\varepsilon_1 = 1.11\%$, 3.33%, 5.56% and 7.78%, corresponding to frequency of 279, 278, 274 and 272 Hz, respectively”

Fig 5c. What are the different curves?

We thank the reviewer's comment. Fig. 5c shows the transmission spectra under different strains. The colors of these curves correspond to symbol colors in Fig. 5b. We revised the manuscript and figure caption of Fig. 5 to make it clear.

The revised manuscript in page 9 paragraph 1 is attached as below:

“Fig. 5c presents the simulated transmission spectra of transverse wave for the four selected strain levels, and all of them have sharp transmission peaks. The transmission peak shifts to lower frequency or higher frequency when the metamaterial is stretched or compressed.”

The revised figure caption of Fig. 5 is attached as below:

“c, The simulated transverse wave transmission peaks at four selected strains $\varepsilon_1 = 1.11\%$, 3.33%, 5.56%, 7.78%, corresponding to the four colored symbols marked in b.”

Fig 5d and e. What means “sequence”? Where the displacements are measured?

We thank the reviewer's comment. The sequence numbers we defined are shown in the middle panel of Fig. 4b. In experiment, we inserted the accelerometer into the holes indicated by cyan line in Fig. 4b, to detect displacement. In order to illustrate the displacement field, we use this parameter as the index.

Are we talking about longitudinal or lateral displacements? Where is the interface?

We thank the reviewer's comment. In our case, we have a transverse wave band gap between the first band and the second band, so the displacement fields we are discussing are vertical (lateral). The position of interface is shown in Fig. 5a, which is the boundary of one of the metamaterials. We revised the manuscript by emphasizing that the polarization of displacement field is vertical.

The author could draw a horizontal line in Fig 5a to show where it is measured.

We thank the reviewer's kind suggestion. We have marked the measured position in Fig. 4b and Supplementary Fig. 7. Therefore, in terms of the concision of paper, we added a sentence in the main text to indicate that the measured position is the same as the former one.

The revised manuscript in page 9 paragraph 1 is attached as below:

“Corresponding experimental displacement field distributions are displayed in Fig. 5d to confirm the existence of topological interface states using the experimental setup detailed in Supplementary Fig. 7.”

The author should also draw a vertical line in Fig 5d and e showing the interface position.

Good suggestion! We revised the Fig. 5d and Fig. 5e with a black dashed line to indicate the interface position and added a sentence in figure caption of Fig. 5 to make it clear *“The black dashed lines in a, d, e indicate the position of interface between two metamaterials”*.

Fig 5f should be an inset of Fig 5a?

We thank the reviewer's comment. Fig. 5a indicates our strategy and design to achieve tunable topological states and Fig. 5f suggests experimental setup of dynamical manipulation of topological states. The two aspects are the logically progressive relationship. It is reasonable to keep them separated.

Fig 5b and 5c should be exchanged.

We thank the reviewer's comments. Fig. 5b illustrates the prediction of frequency shift of interface state based on the projected band structure calculation. Fig. 5c demonstrates tunable capability of the high transmission peak that interface state arises. We think that the prediction is supposed to be prior to phenomenon, which is better to understand.

After three careful readings, I do not still even understand what Fig 5 is supposed to demonstrate.

Fig. 5 corresponds to session of dynamical manipulation of topological interface states. Fig. 5a presents our strategy to realize the tunable topological interface state. We fix one metamaterials in compressive state and stretch or compress the other one. Fig. 5b shows the numerical calculation of the frequency of interface state as a function of strain. It shows that the frequency will decrease as the strain increases. Fig. 5c illustrates the transmission spectra for four strain levels, where the sharp transmission peaks arise. The frequencies of the transmission peaks agree with the prediction in Fig. 5b. Fig. 5d presents the displacement fields in corresponding frequencies, indicating topological interface state. Fig. 5e indicates displacement fields under different frequency along the dashed line in Fig. 5b. It shows that under certain frequency does interface state arise (274 Hz in our case). Fig. 5f shows the experiment setup of dynamical manipulation of topological interface states. When we fix the excitation frequency and deform the metamaterials repeatedly, the topological interface state can appear and disappear. We revised this session carefully to make it clear.

I do understand the difficulty of realizing the beam splitter in Fig 4. The efficiency is close to 50%. Do the author have an idea if this is due to a limitation of the material (dissipation?), or a finite size effect etc...? Or is there any fundamental limit?

We thank the reviewer's comment on beam splitter. From the simulation result (Fig. 4b), the beam splitting is nearly perfect. However, limited by experiment setup, the result is not perfect even though we can still observe the beam-splitting phenomenon. We analyzed and summarized the possible reasons. The difference between simulations and experiments may contribute to two reasons. First, the detection position where we put the accelerometer may affect the magnitude of the displacement. The vibration is mainly concentrated on the matrix rather than around the holes, so the measured displacement is smaller than the simulation results. Second, the fixtures we put on the two edges of the interface of the elastomer sample may affect the detection of the displacement. On the one hand, the fixtures constrict the vibration. On the other hand, there exists a tiny intermediate region between two metamaterials, which has an effect on the measurement. We revised our manuscript by adding the discussion.

The revised manuscript in page 8 paragraph 1 is attached as below:

“First, the detection position where we put the accelerometer may affect the magnitude of the displacement. The vibration is mainly concentrated on the matrix rather than around the holes, so the measured displacement is slightly smaller than the simulation results. Second, the fixtures we put on the two edges of the interface line between two metamaterials may constrict the vibration at the vicinity of the clippers. In addition, a tiny intermediate region between two metamaterials arisen from the different strains may also affect the measurement of the displacement.”

The author should discuss (or at least mention) the relevance of their work in the framework of very recent experimental realization. Especially

For the beam splitter part:

Topological sound in active-liquid metamaterials, Souslov *et al*, *Nature physics* (2017)

Is there any relation between what these authors called circulation and your system which could explain the direction of propagation of sound

For the design:

Programmable Mechanical Metamaterials. Florijn *et al*, *Phys. Rev. Lett.* 113, 175503 (2014)

For the topological interface

Selective buckling via states of self-stress in topological metamaterials, Paulose *et al.*, *Proceedings of the national academy of sciences* (2015)

We thank the reviewer's recommendation on literatures. These literatures are helpful and instructive. The first literature provides a smart design to break time-reversal symmetry using spontaneously flowing active fluid. Actually, it is the simulation realization, not the experimental realization as reviewer mentioned. Since our results are based on time-reversal preserving system and can be explained as difference of the group velocity shown in Supplementary Figure 6, there is no certain relation with spin-locked chiral propagation of sound in time-reversal breaking system. We revised the manuscript by adding the discussion.

The revised manuscript in page 8 paragraph 1 is attached as below:

“This feature may find application in phonon frequency splitter due to different group velocities as presented in the projected band structure (Supplementary Fig. 6), which differs from the chiral propagation in time-reversal breaking system³⁵.”

The second literature provides an impressive strategy to create programmable mechanical metamaterials with mechanical confinement. Indeed, the novel mechanical response may inspire soft elastic metamaterials design in the future. In addition, the topological properties of the programmable mechanical metamaterial are worthwhile to pursue. We revised the manuscript by citing this literature.

The revised manuscript in page 3 paragraph 1 is attached as below:

“Besides, the programmable mechanical behaviors have been achieved in a mechanical metamaterial, which may inspire new tunable devices²⁵.”

The third literature provides the creation of metamaterials that display reliable adaptive behavior controlled by external mechanical loads, with the dual partners of the floppy modes and the states of self-stress. The interesting zero-frequency properties are useful for structural properties and thermodynamic quantities, but they cannot reveal the transport of elastic waves directly. If we would like to achieve it, we have to address the high-frequency properties, which are our case. We also included this literature into our references and revised the manuscript.

The revised manuscript in page 8 paragraph 2 is attached as below:

“The zero-frequency adaptive behavior controlled by external mechanical loads has been displayed, with floppy modes and the states of self-stress³⁶.”

Minor concerns

All the caption should be expanded. The colors, symbol, if it experimental or numerical (etc...) should appear in the caption.

We thank the reviewer’s helpful suggestion. We expanded all our figure captions to make figures clear.

Abstract, line 18. “arbitrary”, please rephrase.

We thank the reviewer’s kind suggestion. We deleted the term “arbitrary”. We rephrased the sentence “...we further demonstrate the formation and dynamical tunability of topological interface states by mechanical deformation and the manipulation of elastic wave propagation.” We also rephrased the sentence in Page 3 Paragraph 1 “...as well as manipulation of elastic wave propagation.”

Page 2 line 45 “Elastic waves,....., have”

We thank the reviewer’s kind suggestion. We rephrased the sentence “Elastic waves, also known as small oscillations in solids, have potential applications in information carrying¹⁹ as well as seismic monitoring²⁰”.

Page 2 line 48 “which is ... prevention” It is overstated no? It could be useful, but at the current stage, it is far for real earthquake prevention.

We thank the reviewer's comment. Indeed, the elastic metamaterials have application on vibration isolation, but as the reviewer have said, it is far from practical operation in real earthquake. Therefore, we rephrased the sentence "*Through creating band gaps in architected materials with periodic porous structures, elastic wave can be dramatically attenuated, which is particularly useful in vibration isolation²¹*".

Page 2 line 52 "advanced fabrication technique", please give some concrete example.

We thank the reviewer's good suggestion. We gave a specific example such as directional solidification to make mechanically anisotropic.

The revised manuscript in page 2 paragraph 2 is attached as below:

"Thanks to the development of advanced fabrication technique, such as directional solidification..."

Page 5 line 144 'Put together', please rephrase

We thank the reviewer's comment. We replaced the "put together" with "edge-to-edge joint" as shown in page 6 paragraph 2: "*When two elastic systems with different topological invariants are edge-to-edge joint, the topological interface states can be predicted to emerge according to the surface impedance match condition $Z_L(\omega, k_{||}) + Z_R(\omega, k_{||}) = 0$* ".

Page 7 line 192-194. Please rephrase. Why is there a criterion 0.9-1.1. It seems a bit arbitrary.

We thank the reviewer's comment. It is reasonable to set the intermediate band, because there exists an intermediate region between two metamaterials in experiment rather than the abrupt change in simulation. The frequency boundary of left propagation mode and right propagation mode is less obvious than that in simulation. Therefore, we choose the area with the most obvious change along the ratio curve, whose lower ratio is about 0.9 and upper ratio is about 1.1. We revised the manuscript by adding explanations for defining the intermediate band.

The revised manuscript in page 8 paragraph 1 is attached as below:

"We define the area of the displacement ratio between 0.9 ~ 1.1 as intermediate mode (grey region in Fig. 4d), which corresponds to the most abrupt area in ratio curve in the frequency range of 293 ~ 295 Hz."

Page 7 line 206-208 Please rephrase. After three careful reading, I still do not understand what the authors means.

We thank the reviewer's comment. We rewrote the sentence to make it clear.

The revised manuscript in page 8 paragraph 2 is attached as below:

"According to the topological phase diagram (Fig. 2b), the metamaterial (0.68, -8.89%) at right side has a relatively large band gap in frequency of 263 ~ 298 Hz. While the metamaterial (0.6, ϵ_1) at left side has a common gap with that at right side when ϵ_1 changes in the range of -1.11% ~ 8.89%."

Part “dynamical manipulation of topological interface”. I do not know why but this part is poorly written and not well organized. There is back and forth between figures (5a 5b 5c 5d 5b 5e 5f 4a 5b 4b) which make it extremely hard to understand. After careful readings, I still do not understand what Fig5 is supposed to demonstrate.

We thank the reviewer’s comment on clarity of session “dynamical manipulation of topological interface state”. Since we would like to show the dynamical manipulation of topological interface state, we deleted the fourth and fifth paragraphs of this session that refers to the beam splitter. Besides, we rewrote this part and made it readable.

Page 7 line 208 “Obviously” Please remove. Nothing is a priori obvious.

We thank the reviewer’s suggestion. We removed the term “obviously”.

Page 8 line 216 “at four strain...” it should “for four ...”.

We replaced the “at four strain levels” with “for four strain levels”. The revised sentence is “*Fig. 5c presents the simulated transmission spectra of transverse wave for the four selected strain levels ...*”

Page 8 231-232 “Actually.... ” What mean the author mean? Please rephrase

We thank the reviewer’s comment. In order to make this session concise and focus on the dynamical manipulation of topological interface state, we deleted the fourth and fifth paragraphs of this session that refers to the beam splitter.

Page 8 line 235 “transport along the interface”. Please rephrase as one could understand you are talking about edge states (which is not the case here).

We thank the reviewer’s comment. In order to make this session concise and focus on the dynamical manipulation of topological interface state, we deleted the fourth and fifth paragraphs of this session that refers to the beam splitter.

Page 9 line 268 “potential in military and communication field”. Please give some examples, otherwise this sentence is purposeless. I would not have brought into light military applications, but on this particular point I must admit that it is only my personal opinion and taste.

We thank the reviewer’s useful suggestion. Since we have claimed some specific applications in the context, we do not think that it is necessary to declare the general applications here, so we deleted this sentence.

Movie What it is the purpose of the movie? It shows a sort of fluctuation of a signal on oscilloscope which measured an acceleration. It seems that there is an upper threshold, but I have no idea what it should refer to. The author should make an effort of presentation

We thank the reviewer's comment on clarity of Supplementary Movie. We added several explanations in the main text and made it clear as much as possible. We also added additional marks in the movie to show the appearance and disappearance for the acceleration signals.

The revised manuscript is attached as below:

“Fig. 5f shows a snapshot of the movie, indicating the experimental setup and dynamical process. When the input vibration is fixed at frequency of 274 Hz, the topological interface states emerge at the strain of 5.56%; the interface states disappear when the strain is larger or smaller than 5.56%. The movie shows a time dependent acceleration signal when we stretch or compress the metamaterial. The appearance of the topological interface states is observed when the acceleration value reaches the maximum, which is marked in the Supplementary movie.”

Supplementary Fig 2b (upper colorplot). The colorplot does not present a top/down symmetry. Why is this?

We thank the reviewer's comment. Since the system at P and Q points possess σ_x mirror symmetry, the real part of eigen field should be either an odd or even function of x . Therefore, there is nothing to do with the field distribution along y direction.

Reviewer #4

In this paper, authors present an experimental observation with numerical simulation of tunable topological states in soft elastic metamaterials. They show that they can switch the topological phase by changing filling ratio, tension or compression of the elastic metamaterials. To convince readers they provide a topological phase diagram of elastic metamaterials under deformation.

We thank the reviewer's positive comments on our work. Following the reviewer's suggestion, we have carefully revised the manuscript.

Even if the paper looks well written in the introduction, I think it is difficult to deeply understand scientific arguments of the authors. Very often one has the impression that it is somehow as the authors are saying “can be formed accidentally”. I would avoid such statement, as the content of the paper is indeed very technical and not so easy to obtain. It give the impression that the main strategy, if it exist, is “fulling around”. I think the paper should be improved significantly and then should be publishable even in Nature Commun
...

We thank the reviewer's comment on the adequacy of statement and encouragement on publishing. The Dirac cone can be degenerated either accidentally or deterministically. By deliberately adjusting the parameter, the Dirac cone can be formed, which is called accidental degeneracy. Since this statement of “Dirac cone can be formed accidentally” may be misunderstood, we revised the sentence “*a twofold Dirac cone at M point can be formed by accidental degeneracy, as presented in Fig. 1c.*”

Now technically, there is a mystery to me, how they connect the hyperelastic model to their structures. It is not enough to know the elastic law. Under a constrain, not all contributions in a microstructure will be “deformed” by the same way. I would recommend to the authors to read the work of William Parnel (Univ of Manchester) who deeply studied this effect.

We thank the reviewer’s question on application of hyperelastic model and suggestion on reading some literatures of William Parnel. First, the hyperelastic model is an indispensable way to describe the deformation of soft materials. In order to model the soft elastic metamaterial accurately, we test the soft material experimentally and fit the experiment data using hyperelastic model. From the nearly perfect match between simulation and experiment results, we think the application of hyperelastic model is essential. Second, these literatures of William Parnel are helpful and instructive. His work focuses on understanding and developing the model of soft phononic crystals, which is usually associated with large strain. Besides, the literatures provide the method that the reference reciprocal space and current reciprocal space are related by an affine mapping, which is exactly what we use in our numerical simulation. As we mentioned in our manuscript, strain in our case is small that we can consider the deformation of every unit cell as a uniform deformation. The geometry changes are shown in Supplementary Fig. 1.

For the numerical model, it is not clear to me what they have really done. I have worked with Comsol for 10 years and such study is far from been trivial. I would appreciate some more details in this part. How they deal with overall constrain and with the post linearization of the problem.

We thank the reviewer’s comment. We are glad to tell the reviewer more details about our numerical simulation. Numerical simulations were performed by using COMSOL Multiphysics and were implemented in the two-dimensional acoustic-structure coupling module. First, we investigated the deformation of our soft elastic metamaterial by applying boundary constraint and displacement on the model. The applied strain in our study was relatively small ($< 9\%$) so we assume the unit cell experiencing a uniform deformation. Thus, we obtained the data of deformed hexagonal unit cell as shown in Supplementary Fig. 1. Second, we calculated the band structure and eigen mode of deformed unit cell with corresponding periodic condition to satisfy current reciprocal space. As for the transmission calculation, we used a harmonic force to excite the model and collected the transmitted signal on the outgoing end. Since the soft elastic metamaterial experiences a small deformation, the observed geometry changes of unit cell are nearly linear, which are indicated by the plots of five parameters as a function of strain (Supplementary Fig. 1). Thus, the high order nonlinear term of strain is ignored in our numerical simulation. We revised the Method session and made it clear.

The revised manuscript is attached as below:

“Numerical simulations. Numerical simulations are performed by using COMSOL Multiphysics, a finite-element analysis and solver software. The simulations are implemented in the two-dimensional acoustic-structure coupling module including the actual geometric size and relevant properties. The system consists of the air resonator (vacuum and water) and soft material Ecoflex 0030. The parameters used for air are mass density 1.29 kg/m^3 and sound speed 340 m/s . The mechanical properties for Ecoflex

0030 are extracted from Yeoh hyperelastic model fit from experiment data. The geometric parameters of unit cell are calculated as the elastic metamaterial is under small mechanical deformation ($< 9\%$) (Supplementary Fig. 1). The deformed unit cell with periodic boundary conditions is used to calculate the band structure and eigen mode, while the supercell with unidirectional periodic condition is used to calculate the projected band structure. For calculating the transmission spectra, an external force with different frequencies is imposed on the boundary of finite size sample to mimic the incident wave. In addition, the displacement on the outgoing end is collected as the transmitted signal. The data line is set two to three lattice constants in front of the edge of PML depending on the frequency of input. For the calculation of wave propagation in finite sample, the PMLs are set around the sample to prevent the leakage of energy.”

In figure 1, what is longitude in “Dash lines indicate longitude wave bands”. Please explain how you get or define the Zak phase.

We thank the reviewer’s comment on inaccurate statement on longitudinal wave. Longitudinal waves are waves that the displacement of medium is parallel to the direction of propagation of waves. The Zak phase is a Berry phase across an appropriately chosen one-dimensional Brillouin zone that is PQ direction with a fixed $k_{||}$ in our case. The reviewer can find out the details in the Supplementary note 1 and Supplementary figure 2. We replaced the “longitude” with “longitudinal” in figure caption of Fig. 1 and main text.

Figure 2 is nicely colourful but what is the meaning of each colour? Could you add a legend to the colours?

We thank the reviewer’s comments. Yellow represents the band gap with $\zeta < 0$ and cyan represents the band gap with $\zeta > 0$. Colors are to differ distinct topological phase which is determined by the band gap sign. Actually, we have added the symbols to this diagram with black font showing the meaning of corresponding colored zones.

Figure 3, please specify the source term and its real position. Generally it is a bad idea to have a source or recording term on the edge with a PML. It always depends on the definition of the PML, which in mechanics can be less efficient than in electromagnetism.

We thank the reviewer’s suggestion on specifying the input port and data line. An external harmonic force with different angles is exerted on the left side to provide an input of elastic waves. A perfect matched layer (PML) is added at the right side and periodic boundary conditions are imposed on the upper and lower edges. Surely, it is not a good idea to set the data line on the edge of PML as the reviewer said. In the simulation, the data line is set two to three lattice constants in front of the edge of PML depending on the frequency of input elastic wave. Following the reviewer’s suggestion, we revised the Method session in our manuscript.

The revised manuscript is attached as below:

“For calculating the transmission spectra, an external force with different frequencies is imposed on the boundary of finite size sample to mimic the incident wave. In addition, the displacement on the outgoing end is collected as the transmitted signal. The data line is

set two to three lattice constants in front of the edge of PML depending on the frequency of input.”

What is the bar on top of Γ and M in the panel b?

We understand the concern of the reviewer. It is the common practice in calculation of projected band structure to write a bar above Γ and M, which distinguishes the Γ and M from band structure of unit cell. (*Phys. Rev. B* **90**, 075423 (2014))

We revised the figure caption of Fig. 3 by rewriting the sentence “*The simulated projected band structure along k_x direction is indicated by $\bar{\Gamma}\bar{M}$ with transverse interface modes. The bars above Γ and M are used to distinguish the Γ and M from band structure of unit cell*”.

Figure 4, be more precise in the caption. I have no clue how this was done nor what is really shown by reading the caption.

We thank the reviewer’s suggestion. Following the reviewer’s suggestion, we extended the figure caption and made it clear.

The revised figure caption is attached as below:

“Figure 4 | Experimental observation of topological interface state and demonstration of elastic wave splitter. a, Experimental setup of a Ecoflex slab consisting of two elastic metamaterials, (0.6, 5.56%|0.68, -8.89%). Magenta, red and blue lines schematically indicate elastic waves propagation directions as a function of input frequency. **b**, Numerical simulations of vertical displacement fields with different transverse wave propagations at three input frequencies: 261 Hz, 274 Hz and 286 Hz, from top to bottom. **c**, Experimental observation of topological interface state at 274 Hz corresponding to the second panel in **b**, by measuring the displacement of the 24 holes marked by cyan line. Values of the measured displacements represent the mean of n tests ($n = 5$). Sequence numbers marked in red indicate the hole numbers. The black dashed line indicates the position of interface between two metamaterials. **d**, Experimentally measured displacement on the right side at the magenta hole is presented by the magenta dashed curve. Experimentally measured displacement on the left side at the blue hole is presented by the blue dashed curve. The displacement ratio of the left side over the right side, L/R is presented as a function of input frequency (black solid curve). Magenta domain indicates the right propagation mode, while the blue domain reveals the left propagation mode. Grey region is the intermediate mode defined in main text.”

Figure 5, I do not understand the panel (a). What is the physical origin of the “interface”? what are the picks in the panel c? that looks as single point picks which is not acceptable as measurement from the resolution point of view.

We thank the reviewer’s questions. The term “interface” means boundary. In our case, the interface means the junction line of two metamaterials, where we drew a black dashed line to indicate the position. Fig. 5c presents the evaluation of transmission spectra from simulation for four selected strain levels ($\epsilon_1 = 1.11\%$, 3.33%, 5.56%, 7.78%). As shown in Fig. 3b, there is a flat band in the band gap, so the corresponding spectrum presents transmission peak of transverse wave with one certain frequency. Therefore, in terms of

the experiment, it is difficult to measure a single peak in the transmission spectra. We also revised the figure caption of Fig. 5 to make it clear.

The revised figure caption of Fig. 5 is attached as below:

“Figure 5 | Tunability of soft topological system. a, The schematic of tunable topological system with two metamaterials, $(0.6, \epsilon_1|0.68, -8.89\%)$, where strain ϵ_1 is variable in the range of $-5.56\% \sim 8.89\%$. **b**, The numerically simulated frequencies of topological interface state as a function of strain. The topological state emerges at a certain frequency and the frequency decreases as the strain increases. **c**, The simulated transverse wave transmission peaks for four selected strains $\epsilon_1 = 1.11\%, 3.33\%, 5.56\%, 7.78\%$, corresponding to the four colored symbols marked in **b**. **d**, The experimentally measured vertical displacement field distributions at the same four selected strains. The markers and colors have their correspondences in **b**. **e**, The experimentally measured displacement field distributions at five selected frequencies along the purple dashed line in **b**, 272 Hz, 273 Hz, 274 Hz, 275 Hz and 276 Hz. **f**, A snapshot of experimental demonstration of dynamical manipulation of topological interface state. The black dashed lines in **a**, **d**, **e** indicate the position of interface between two metamaterials.”

Reviewers' comments:

Reviewer #1 (Remarks to the Author):

The authors have greatly improved the clarity of the manuscript. Except as otherwise noted, they have addressed my concerns. In particular, Fig. 5e seems to demonstrate that the experimental system possesses the boundary mode predicted by numerical simulations, and the analytic theory indicates that it is topological. It might be nice to have a similar frequency sweep at a strain level which would not have a topological mode, but given that the topological mode appears sharply at the frequency predicted this doesn't seem necessary.

However, I feel that the authors should more carefully delineate how their work differs from past results in this area. The statement in the abstract that "Theoretical modeling and numerical simulation of elastic topological states have been reported whereas the experimental observation remains unexplored", coupled with the authors' reports of their own results gives me the impression that elastic topological states have not previously been experimentally observed. I don't think the authors mean to give that impression, since they cite works such as Pal and Ruzzene, so I would suggest changing "unexplored" to "relatively unexplored" or something of that nature.

I would also like the authors to address here, and consider addressing in the paper, the two papers I mentioned (Chen Upadhyaya Vitelli 2014 and Rocklin Zhou Sun Mao 2017). The manuscript currently may be read as being the only work to attempt to create a tunable topological state in a soft, elastic material. Both of the mentioned works attempt to do just that, in very different contexts than the present work. Those systems are not elastically stable, so that deforming them to tune the states does not require stress, and the topological modes occur at zero frequency (and the experiments were quite rudimentary). But they do seem to have demonstrated tuning of topological states in soft, elastic systems.

One way to resolve this would be to reference those papers and note some or all of the ways in which they differ from the elastic waves studied in the present work, much as the authors did in referencing Souslov et al. following the comments of another referee. Another way to resolve this would be to tighten the characterizations of the present work so that they would not overlap with the other papers, for example by replacing references to "topological states" to references to "topological waves". Alternately, the authors could explain why they think the paper is clear on this issue in its present form.

Reviewer #2 (Remarks to the Author):

On the whole, I find the changes to the manuscript to be rather minor, and my objections have not been adequately addressed. As a result, I can not recommend the article for publication. More details on each of my original points is below.

1) I still see numerous grammatical errors - indeed it appears as if almost nothing has been fixed. Perhaps it would be beneficial for the authors to hire a copy editor to revise their manuscript?

As far as I can tell, no modifications have been made to the movie, and I'm still not entirely sure what it is supposed to demonstrate.

2a) The discussion of the effect of the clamping is insufficient; if the author's rebuttal is true (that it has no significant effect), then this should be possible to demonstrate with simulations. The authors also state in their rebuttal: "Therefore, according to the perfect match between simulation and experiment, we believe that the topological states still exist." I don't see how they determined it's perfect, since there are no plots that allow me to make a direct comparison (not to mention the poor quality of the data). It is also irrelevant what the author's believe to be true; they need to demonstrate it with their data. In my view, this has not been done. This is especially true with regards to the beamsplitter: if it is truly topological in nature, it should be effective irrespective of imperfections! Indeed, the low level of splitting seen in their data could easily be obtained by regular (non-topological) resonances in either side of the matrix.

An additional way to confirm the topological nature would be the strong suppression of transmission demonstrated in their simulations. Unfortunately the authors have not been able to do this experiment.

2b) How did the authors arrive at the density of the accelerometer? It seems absurdly low, and in any case the density would only be the relevant figure if it were the same thickness as the sample. Instead, I see a cable hanging out of it (which certainly has higher density). In this case, isn't the relevant figure the total mass, including the moving part of the wire, as compared to the hole it replaces? I do not find it credible that this is over 100x lower than the mass of the hole, and if this is indeed the case they need to provide details.

Moreover, the authors state the vibrational modes are concentrated not in the holes, but in the bulk. In this case, might they not be sensitive to even a very tiny bit of added mass (as in this case the mode would only need a very small modification to eliminate the vibration entirely from the mode)? Although it is unfortunate that the authors do not have access to more sophisticated apparatus, this does not excuse questionable data. Cameras capable of recording ~ 1000 frames per second should not be prohibitively expensive, and would provide dramatically better data. (This is especially true if the vibration is concentrated away from the hole - painting dots on top of the lattice and tracking their motion would be trivial.) Alternatively, conducting a simulation with added mass in the holes (including the cable) would provide much more convincing evidence that their technique does not affect the results. At present, I do not think it meets the standard for publishable data, and even if it did I don't think it adequately demonstrates the effects that they claim it does.

In summary, serious textual revisions and new experimental/simulation data is required before I would recommend this manuscript for publication. It is unfortunate that the authors chose not to fix these serious problems, as I do think the research has many commendable aspects.

Reviewer #4 (Remarks to the Author):

Authors have implemented the majority of my concerns except the question on the implementation of the hyperelastic material.

I would advise them to explain it deeper and to add the coupled equations that they solve.

Reviewer #1 (Remarks to the Author):

The authors have greatly improved the clarity of the manuscript. Except as otherwise noted, they have addressed my concerns. In particular, Fig. 5e seems to demonstrate that the experimental system possesses the boundary mode predicted by numerical simulations, and the analytic theory indicates that it is topological. It might be nice to have a similar frequency sweep at a strain level which would not have a topological mode, but given that the topological mode appears sharply at the frequency predicted this doesn't seem necessary.

We thank the reviewer's positive comment. We agree with the reviewer that the topological mode appears sharply at 274 Hz only when we swept the frequency in a range of 272-276 Hz at the strain of 5.56%.

However, I feel that the authors should more carefully delineate how their work differs from past results in this area. The statement in the abstract that "Theoretical modeling and numerical simulation of elastic topological states have been reported whereas the experimental observation remains unexplored", coupled with the authors' reports of their own results gives me the impression that elastic topological states have not previously been experimentally observed. I don't think the authors mean to give that impression, since they cite works such as Pal and Ruzzene, so I would suggest changing "unexplored" to "relatively unexplored" or something of that nature.

We thank the reviewer's kind suggestion. Pal and Ruzzene have theoretically and numerically demonstrated the topological edge wave in discrete solid and continuous plate, but they did not carry out experimental exploration. Since Huber's work has experimentally demonstrated topologically protected wave transport in discrete solid (a combination of springs and levers connected to linear pendulums), we could claim that the elastic topological states have not been experimentally explored in real materials. Therefore, we revised the manuscript by changing "unexplored" to "relatively unexplored".

I would also like the authors to address here, and consider addressing in the paper, the two papers I mentioned (Chen Upadhyaya Vitelli 2014 and Rocklin Zhou Sun Mao 2017). The manuscript currently may be read as being the only work to attempt to create a tunable topological state in a soft, elastic material. Both of the mentioned works attempt to do just that, in very different contexts than the present work. Those systems are not elastically stable, so that deforming them to tune the states does not require stress, and the topological modes occur at zero frequency (and the experiments were quite rudimentary). But they do seem to have demonstrated tuning of topological states in soft, elastic systems.

We thank the reviewer's comment. Firstly, Vitelli's and Mao's works focused on the topological zero-energy motions whose properties are useful for structural properties and thermodynamic quantities, but they cannot reveal the transport of elastic waves directly. Secondly, their focus was on the Maxwell framework consisting of rods and hinges, but it could hardly be recognized as the continuous materials. We proposed the topological high-frequency properties and showed the

dynamical tunability in real materials.

One way to resolve this would be to reference those papers and note some or all of the ways in which they differ from the elastic waves studied in the present work, much as the authors did in referencing Souslov et al. following the comments of another referee. Another way to resolve this would be to tighten the characterizations of the present work so that they would not overlap with the other papers, for example by replacing references to "topological states" to references to "topological waves". Alternately, the authors could explain why they think the paper is clear on this issue in its present form.

We thank the reviewer's helpful solutions. We referred to Vitelli's and Mao's works and discussed the significance of their works. We also added the difference between their and our works and emphasized the characterization of our work.

The revised manuscript in page 3 paragraph 1 is shown as below:

"The tunable topological zero-energy motions based on Maxwell framework consisting of rods and hinges have been put forward. While they have the impact on novel machines and robots, the transport of elastic waves has not yet been directly revealed and the framework can hardly be considered as the continuous medium^{28,29}. The combination of soft materials with high-frequency topological states offers unprecedented opportunities, which requires insight exploration."

Reviewer #2 (Remarks to the Author):

On the whole, I find the changes to the manuscript to be rather minor, and my objections have not been adequately addressed. As a result, I can not recommend the article for publication. More details on each of my original points is below.

We thank the reviewer's comments on our research. Actually, we have revised the manuscript according to the reviewers' comments and suggestions. In the following, we address the reviewer's concerns point by point and revise our manuscript.

1) I still see numerous grammatical errors - indeed it appears as if almost nothing has been fixed. Perhaps it would be beneficial for the authors to hire a copy editor to revise their manuscript?

We thank the reviewer's comments on our manuscript. Following the reviewer's suggestion, we carefully revised all the grammatical errors.

As far as I can tell, no modifications have been made to the movie, and I'm still not entirely sure what it is supposed to demonstrate.

We thank the reviewer's comment on clarity of Supplementary Movie. Actually, we have added

explanations in the manuscript (paragraph 2 in page 9) and additional marks (interface state on/off) in the movie to show the appearance and disappearance of the interface state (Fig. R1). We would like to demonstrate the dynamically tunable topological interface state in our soft elastic metamaterials. We compressed the soft metamaterial with $d/R = 0.68$ to strain $\varepsilon = -8.89\%$ and fixed it. The accelerometer was put into the hole at interface (Number 12). The other part $d/R = 0.6$ was stretched and compressed repeatedly while the vibration (274 Hz) was generated constantly. The movie presented a time-dependent acceleration signal showing the fluctuation of acceleration, further suggesting the fluctuation of displacement. When the displacement reaches the maximum, the interface state is considered to appear.

Figure R1. The snapshot of the time dependent acceleration signals shows the notes of “interface state on” when the displacement reaches the maximum (left panel), otherwise showing “interface state off” (right panel).

2a) The discussion of the effect of the clamping is insufficient; if the author's rebuttal is true (that it has no significant effect), than this should be possible to demonstrate with simulations.

We thank the reviewer’s comment. We numerically simulated the clamping situation. We set a boundary constraint to the corresponding position where the clamps locate and a harmonic force at the edge of the interface. The simulation model with the actual geometry after mechanical deformation was used. As presented in Supplementary Fig. 7b, the resultant simulation results show that the transverse modes are located in the interface region, which is consistent with the simulation results in Fig. 4b and Supplementary Fig. 7a. Thus, the clamping has no significant effect on the generation of the topological states.

The revised supplementary note 2 is shown as below:

“Note that the two blue clips in Supplementary Fig. 8 could generate a small strain intermediate region, as shown in Supplementary Fig. 7b. We numerically simulated the clamping situation by setting a boundary constraint to the corresponding position where the clamps locate and a harmonic force at the edge of the interface. The simulation model with the actual geometry after mechanical deformation was used. As presented in Supplementary Fig. 7b, the resultant simulation results show that the transverse modes are located in the interface region, which is consistent with the simulation results in Fig. 4b and Supplementary Fig. 7a. Thus, the clamping has no significant effect on the generation of the topological states.”

The authors also state in their rebuttal: "Therefore, according to the perfect match between simulation and experiment, we believe that the topological states still exist." I don't see how they determined it's perfect, since there are no plots that allow me to make a direct comparison (not to mention the poor quality of the data). It is also irrelevant what the author's believe to be true; they need to demonstrate it with their data. In my view, this has not been done.

We thank the reviewer's comment. First, when the frequency of excitation signal is 274 Hz that is the simulated frequency of topological interface state, the detected displacements at 24 holes are summarized in Fig. 4c. The typical confined displacement field is shown. The vibration at the left of the interface drops slower than that at the right, which is consistent with the field distribution in Fig. 4b. Second, when the frequency of excitation signal deviates from 274 Hz, the topological interface state disappears sharply as shown in Fig. 5e. Therefore, we believe that the experiment and simulation agree well.

This is especially true with regards to the beamsplitter: if it is truly topological in nature, it should be effective irrespective of imperfections! Indeed, the low level of splitting seen in their data could easily be obtained by regular (non-topological) resonances in either side of the matrix. An additional way to confirm the topological nature would be the strong suppression of transmission demonstrated in their simulations. Unfortunately the authors have not been able to do this experiment.

We thank the reviewer's comment. Since our elastic system has a band gap along ΓM direction, the problem can be effectively considered as the one-dimensional problem. Generally, we do not discuss the defect-immune characteristic in one-dimensional or quasi-one-dimensional system, which is different from the two- or three- dimensional system. Instead, we usually put it in this way: as long as the inversion symmetry of our system exists, the topological state is stable at its frequency (*Nat. Phys.* **11**, 240–244 (2015), *Sci. Rep.* **6**, 29202 (2016), *Nat. Commun.* **8**, 14587 (2017), *Phys. Rev. B* **90**, 075423 (2014)). Besides, we do not mention in our manuscript that the elastic waves splitting is in the topological way. In Supplementary Figure 6, the red dotted curve represents the topological interface state whose frequency is 274 Hz. At the frequency beyond or below 274 Hz, there are several non-topological discrete bands where the difference of group velocity results in the different propagation direction of elastic waves.

2b) How did the authors arrive at the density of the accelerometer? It seems absurdly low, and in any case the density would only be the relevant figure if it were the same thickness as the sample. Instead, I see a cable hanging out of it (which certainly has higher density). In this case, isn't the relevant figure the total mass, including the moving part of the wire, as compared to the hole it replaces? I do not find it credible that this is over 100x lower than the mass of the hole, and if this is indeed the case they need to provide details.

We thank the reviewer's comment. We recalculated the effective density of our accelerometer according to the product specification. The mass and the volume of the accelerometer are 0.6 g and 142.88 mm³, respectively. Therefore, the effective density of the accelerometer is 4.2 g/cm³.

We revised the type of the density of our sample and it should be 1.03 g/cm^3 . The wire is so flexible that it cannot virtually transfer the vibration. For the simplicity of the simulation, we did not account the wire into our model. The added mass might affect the measured results, so we added the simulation and experiment to confirm that such experimental estimate is reasonable, as shown in the below response.

Moreover, the authors state the vibrational modes are concentrated not in the holes, but in the bulk. In this case, might they not be sensitive to even a very tiny bit of added mass (as in this case the mode would only need a very small modification to eliminate the vibration entirely from the mode)? Although it is unfortunate that the authors do not have access to more sophisticated apparatus, this does not excuse questionable data. Cameras capable of recording ~ 1000 frames per second should not be prohibitively expensive, and would provide dramatically better data. (This is especially true if the vibration is concentrated away from the hole - painting dots on top of the lattice and tracking their motion would be trivial.) Alternatively, conducting a simulation with added mass in the holes (including the cable) would provide much more convincing evidence that their technique does not affect the results. At present, I do not think it meets the standard for publishable data, and even if it did I don't think it adequately demonstrates the effects that they claim it does.

We thank the reviewer's comment. We tried to observe the vibration using the high-speed camera. The propagating elastic waves in our case can only induce small amplitude of vibration ($< 400 \text{ nm}$), which is beyond the resolution of our high-speed camera. However, we put forward the two-accelerometer method to confirm that our experiment technique was valid. We used two accelerometers (A1 and A2): A1 was inserted into the No.13 hole as an obstacle and A2 was inserted into holes (No.1 to No.24) to measure the displacement (Supplementary Figure 7e). The measured displacement field (the blue curve in Supplementary Fig. 7f) is almost the same as the former one (the black curve in Supplementary Fig. 7f) measured by one-accelerometer method (Supplementary Figure 7d). At the same time, we also recorded the data that A1 collected and summarized the variation as a function of measured position of A2. The results showed that the measured displacement (the orange curve in Supplementary Fig. 7f) collected by A1 remained almost stable, suggesting our technique can hardly affect the results.

Besides, we also carried out simulations with added mass. Because the precise structure inside the commercial accelerometer was unknown, we considered the accelerometer as a solid titanium object. As presented in Supplementary Fig. 7c, the resultant simulation results show that the transverse modes are located in the interface region, which is consistent with the simulation results in Fig. 4b and Supplementary Fig. 7a.

Further, we calculated the projected band structure with the added mass, no matter along k_x direction or along k_y direction, the topological mode could be clearly observed (Supplementary Fig. 7g). After analyzing the band structure, we found that the added mass would affect longitudinal mode, such as generating the longitudinal wave band gap and the additional longitudinal mode, but it would not affect transverse mode.

Thus, from the experiments and the simulations, it is convinced that our measurement technique using accelerometer does not affect the results. We added a supplementary figure and revised the manuscript and supplementary note 2.

The supplementary figure 7 is shown as below:

Supplementary Figure 7 | Clamping effect and accelerometer effect. **a.** The simulated topological interface state of $(0.6, 5.56\%|0.68, -8.89\%)$ metamaterial shown in Fig. 4b. **b.** The simulated topological interface state of $(0.6, 5.56\%|0.68, -8.89\%)$ metamaterial with an intermediate region modeled by deformed geometry. **c.** The simulated topological interface state of $(0.6, 5.56\%|0.68, -8.89\%)$ metamaterial with an accelerometer in the number 13 hole. **d.** The schematic of one-accelerometer method to measure the topological interface state. One accelerometer is used to measure the displacement field by detecting the displacement from number 1 hole to number 24 hole. **e.** The schematic of two-accelerometer method to measure the topological interface state and to confirm the tiny effect of the added mass. One accelerometer (A2) is inserted into the holes from number 1 to number 24 to measure the displacement and the other one (A1) is inserted into the number 13 hole as an obstacle. The data from the “obstacle” accelerometer is also recorded. **f.** The comparison between measurements using one-accelerometer

method and two-accelerometer method. The black curve shows the experimental results using one accelerometer. The blue curve shows the experimental results when A1 is placed at the number 13 hole and A2 is used to measure the displacement. Note that the data of number 13 hole is not measured because of the occupation of number 13 hole by A1. The orange curve shows the variation of vibration in interface position when A2 changes the measured hole. **g.** The left panel shows the simulated projected band structure along k_x direction indicated by $\bar{\Gamma}\bar{M}$ with an accelerometer in the number 13 hole. The bars above Γ and M are used to distinguish the Γ and M from band structure of unit cell. Red line indicates interface state independent of bulk state (magenta region) and gray lines indicate longitudinal wave modes. The right panel shows the simulated projected band structure along k_y direction with an accelerometer in the number 13 hole. A topological band is shown near $k_y = 0$ marked in red dots. The blue dots show the longitudinal modes arisen from the added mass.

The revised manuscript is shown as below:

“Considering the stable characteristic of topological interface state, the added mass effect can be neglected, as supported by further simulations and experiments in Supplementary Fig.7.”

The revised supplementary note 2 is shown as below:

“In order to investigate whether the measurement method affects experimental results, we carry out another experiment: one accelerometer is inserted into the holes to measure the displacement and the other one is inserted into the number 13 hole as an obstacle. The data from the “obstacle” accelerometer are also recorded. The measured displacement field (blue curve in Supplementary Fig. 7f) is almost the same as the former one (black curve in Supplementary Fig. 7f). Besides, the displacement at the number 13 hole is nearly stable when the accelerometer moves from number 1 to number 24 hole (orange curve in Supplementary Fig. 7f). The additional experiment suggests our technique can hardly affect the results.”

In summary, serious textual revisions and new experimental/simulation data is required before I would recommend this manuscript for publication. It is unfortunate that the authors chose not to fix these serious problems, as I do think the research has many commendable aspects.

We thank the reviewer’s comment. We have tried our best to improve our manuscript by adding new experimental and simulation data. Besides, we carefully revised the manuscript to make it clear.

Reviewer #4 (Remarks to the Author):

Authors have implemented the majority of my concerns except the question on the implementation of the hyperelastic material. I would advice them to explain it deeper and to add the coupled equations that they solve.

We thank the reviewer's comment. We added a supplementary section on explanation and coupled equations used in the static and wave propagation analysis.

The supplementary note 3 is shown as below:

Supplementary note 3 | Governing equations and wave propagation analysis

We delineate the undeformed and the deformed states by Ω_0 and Ω , and the material and spatial points by \mathbf{X} and \mathbf{x} , respectively. The motion of the material is shown as affine mapping that assigns material points \mathbf{X} to spatial points \mathbf{x} ⁵.

$$x_i = \mathcal{J}_i(\mathbf{X}_I, t) \quad (\text{S1})$$

The deformation gradient of the motion is defined as:

$$F_{iJ} = \frac{\partial x_i}{\partial X_J} \quad (\text{S2})$$

In our case, the local term of linear momentum balance can be expressed in the undeformed configuration as⁶:

$$\frac{\partial P_{iJ}}{\partial X_J} - \rho \frac{D^2 u_i}{Dt^2} = 0 \quad (\text{S3})$$

where \mathbf{P} is the first Piola-Kirchhoff stress, $\mathbf{U} = \mathbf{x} - \mathbf{X}$ is the displacement field. In our case, the hyperelastic model is described by the strain energy density function W . So the first Piola-Kirchhoff stress \mathbf{P} can be written as:

$$P_{iJ} = \frac{\partial W}{\partial F_{iJ}} \quad (\text{S4})$$

Further, the Cauchy stress can be expressed as $\boldsymbol{\sigma} = J^{-1} \mathbf{P} \mathbf{F}^T$, where $J = \det(\mathbf{F})$. Therefore, the equation (S3) can be described spatially:

$$\frac{\partial \sigma_{ij}}{\partial x_j} - \rho \frac{\partial^2 u_i}{\partial t^2} = 0 \quad (\text{S5})$$

Then, we consider a small perturbation superimposed on the given deformed configuration that takes the continuous material to a new equilibrium. The incremental problem can be described in the deformed configuration:

$$\frac{\partial \hat{\sigma}_{ij}}{\partial x_j} - \rho \frac{\partial^2 \hat{u}_i}{\partial t^2} = 0 \quad (\text{S6})$$

Where the bracket symbol on the quantity denotes the increment of the corresponding quantity.

After employing push-forward transformations based on linear momentum⁷, we obtain:

$$\hat{\sigma}_{ij} = J^{-1} \tilde{P}_{ij} F_{jJ} = C_{ijkl} \frac{\partial \tilde{u}_k}{\partial x_l} \quad (\text{S7})$$

Where $C_{ijkl} = J^{-1} F_{jJ} F_{lL} \frac{\partial^2 W}{\partial F_{iI} \partial F_{kK}}$ (S8) represents the spatial elasticity tensor.

Since the amplitude of elastic wave is small, the propagation can be described by

$$\tilde{\mathbf{u}}(\mathbf{x}, t) = \mathbf{u}(\mathbf{x}) e^{-i\omega t} \quad (\text{S8})$$

$$\tilde{\boldsymbol{\sigma}}(\mathbf{x}, t) = \boldsymbol{\sigma}(\mathbf{x}) e^{-i\omega t} \quad (\text{S9})$$

So that equation (6) becomes

$$\nabla \cdot \boldsymbol{\sigma} + \rho \omega^2 \mathbf{u} = 0 \quad (\text{S10})$$

Since our elastic metamaterial is a periodic structure characterized by a unit cell, any periodic function satisfies the condition:

$$\phi(\mathbf{x} + \mathbf{r}) = \phi(\mathbf{x}) \quad (\text{S11})$$

where $\mathbf{r} = r_1 \mathbf{a}_1 + r_2 \mathbf{a}_2$ (S12)

where r_1 and r_2 are arbitrary integers. In deformed configuration, the basic vectors \mathbf{a}_1 and \mathbf{a}_2 are different from the undeformed ones due to the change of reciprocal lattice.

The reciprocal lattice vectors \mathbf{b}_1 and \mathbf{b}_2 can be derived from:

$$\mathbf{a}_i \cdot \mathbf{b}_j = 2\pi \delta_{ij} \quad (\text{S13})$$

Specifically, in 2D lattice:

$$\mathbf{b}_1 = 2\pi \frac{\mathbf{a}_2 \times \mathbf{e}_z}{\mathbf{a}_1 \cdot (\mathbf{a}_2 \times \mathbf{e}_z)} \quad (\text{S14})$$

$$\mathbf{b}_2 = 2\pi \frac{\mathbf{e}_z \times \mathbf{a}_1}{\mathbf{a}_1 \cdot (\mathbf{a}_2 \times \mathbf{e}_z)} \quad (\text{S15})$$

Thus, the reciprocal lattice can be described by:

$$\mathbf{g} = g_1 \mathbf{b}_1 + g_2 \mathbf{b}_2 \quad (\text{S16})$$

Any function $\varphi(\mathbf{k})$ in reciprocal space satisfies⁸:

$$\varphi(\mathbf{k} + \mathbf{g}) = \varphi(\mathbf{k}) \quad (\text{S17})$$

In order to obtain band structures for different strain levels, Bloch boundary condition is applied to the deformed lattice:

$$\mathbf{u}(\mathbf{x} + \mathbf{r}) = \mathbf{u}(\mathbf{x}) e^{i\mathbf{k} \cdot \mathbf{r}} \quad (\text{S18})$$

Then we solve the equation (S10) in the states for different strain levels. Due to the symmetry in equation (S17), we focus on the first Brillouin zone. We calculate the band structure along the path Γ -M-K- Γ shown in Fig. 1c and Fig. 1d.

Reviewers' comments:

Reviewer #1 (Remarks to the Author):

I find that the authors have adequately addressed all of my concerns.

Although one of the other referees raises important points regarding the experimental data, I feel confident in the authors' interpretation of their data based on its combination with simulation and theory.

Having seen the manuscript clarified and the main results justified, I recommend it for publication.

Reviewer #2 (Remarks to the Author):

Again, I do not feel these experimental results are of publishable quality. The supplementary figure with the added mass makes it clear that the presence of the accelerometer DOES significantly affect the results (S. Fig 7a/c), and hence the accelerometer method is simply not an adequate probe of the vibrational states. Every time you place it in a different hole you get a totally different mode! I don't think the anecdotal two accelerometer method proves anything, other than it is insensitive to added mass at that one particular hole. Consequently, I can not recommend publication of any paper based on such a method.

The fact that they were previously unable to correctly calculate the effective mass gives me serious concerns about how careful the authors have been with their results. Additionally, I still see no way to directly compare the experimental results to the theoretical; the authors claim they agree, but they still haven't shown it -- there should be a plot I can directly compare. For example, how can I easily compare fig to fig 4b to 4c? The authors claim in their rebuttal that "The vibration at the left of the interface drops slower than that at the right, which is consistent with the field distribution in Fig. 4b." It is questionable at best if this can even be seen in the data, and in any case this is quite weak evidence.

The beam splitter results continue to be of poor quality, and as far as I can tell from looking at fig 4b, the simulations indicate significantly higher quality results should be obtained. (One presumes this may be due to the effects of added mass.) Moreover, How do any of the experimental results prove a topological mode is observed? I still do not follow their line of reasoning. It seems it should be possible to switch the mode on and off by altering strain -- I guess this is what the video is supposed to show, but by only measuring at a single frequency this allows for the possibility that it has simply changed frequencies. Normally when a paper with both theoretical and experimental results, you measure the same thing in both and show they agree. I do not understand why they have not done this.

Finally, numerous grammatical errors still exist, and the writing is (still) hard to follow in many places. I would encourage the authors to hire a copy editor.

Reviewer #1 (Remarks to the Author):

I find that the authors have adequately addressed all of my concerns. Although one of the other referees raises important points regarding the experimental data, I feel confident in the authors' interpretation of their data based on its combination with simulation and theory. Having seen the manuscript clarified and the main results justified, I recommend it for publication.

We thank the reviewer's positive comments.

Reviewer #2 (Remarks to the Author):

Again, I do not feel these experimental results are of publishable quality. The supplementary figure with the added mass makes it clear that the presence of the accelerometer DOES significantly affect the results (S. Fig 7a/c), and hence the accelerometer method is simply not an adequate probe of the vibrational states. Every time you place it in a different hole you get a totally different mode! I don't think the anecdotal two accelerometer method proves anything, other than it is insensitive to added mass at that one particular hole. Consequently, I can not recommend publication of any paper based on such a method.

We thank the reviewer's comments. The accelerometer method is an effective way to detect vibration and has been widely used in previous works (Adv. Mater. **28**, 5943-5948 (2016), Phys. Rev. Lett. **113**, 014301 (2014)). Specifically, for the stable topological states, Prodan *et al.* measured the Majorana edge states by using accelerometers (*Nat. Commun.* **8**, 14587 (2017)). The microphone whose acoustic impedance is much larger than the air is used in airborne acoustic topological mode analysis as well as reflection phase analysis. Xiao *et al.* obtain the good agreement with experiments and theories (*Nat. Phys.* **11**, 240–244 (2015)).

Furthermore, the two-method-measured displacement fields are almost the same and the displacement at the interface is fluctuating but almost stable. The two-accelerometer method proves that the topological states remain relatively stable when there is an obstacle in the metamaterial. In order to make it clear, we revised the manuscript by adding some discussions.

The revised manuscript in paragraph 3 page 7 is shown as below:

“It is noted that the vibration at the left of the interface drops slower than that at the right, which is consistent with the simulation result in Fig. 4c and the field distribution in Fig. 4b. Note that inserting an accelerometer into the hole to measure the displacement will bring added mass to the sample. The accelerometer method has been employed as an effective way to detect vibration of elastic metamaterials^{21,24}. Considering the stable characteristic of topological interface state, Majorana edge states have already been observed by using accelerometers³⁷. In our case, the further simulations and experiments in Supplementary Fig.7 and Supplementary note 2 confirm

the added mass effect can be neglected so that the estimated displacement field is valid.”

The fact that they were previously unable to correctly calculate the effective mass gives me serious concerns about how careful the authors have been with their results. Additionally, I still see no way to directly compare the experimental results to the theoretical; the authors claim they agree, but they still haven't shown it -- there should be a plot I can directly compare. For example, how can I easily compare fig to fig 4b to 4c? The authors claim in their rebuttal that "The vibration at the left of the interface drops slower than that at the right, which is consistent with the field distribution in Fig. 4b." It is questionable at best if this can even be seen in the data, and in any case this is quite weak evidence.

We thank the reviewer’s comments. In order to show the explicit comparison, we revised the Fig. 4c by adding simulation results. We also added the comparison between experiment results and simulation results of Fig. 5d and Fig. 5e in Supplementary Fig. 8.

The revised Fig. 4c and figure caption are shown as below:

Figure 4 | Experimental observation of topological interface state and demonstration of elastic wave splitter. **a**, Experimental setup of a Ecoflex slab consisting with two elastic metamaterials, (0.6, 5.56%|0.68, -8.89%). Magenta, red and blue lines schematically indicate

elastic waves propagation directions as a function of input frequency. **b**, Numerical simulations of vertical displacement fields with different transverse wave propagations at three input frequencies: 261 Hz, 274 Hz, and 286 Hz, from top to bottom. **c**, Experimental observation of topological interface state at 274 Hz corresponding to the second panel in **b**, by measuring the displacement of the 24 holes marked by cyan line. Values of the measured displacements represent the mean of n tests ($n = 5$). Simulation results are also shown in blue dashed line. Sequence numbers marked in red indicate the hole numbers. The black dashed line indicates the position of interface between two metamaterials. **d**, Experimentally measured displacement on the right side at the magenta hole is presented by the magenta dashed curve. Experimentally measured displacement on the left side at the blue hole is presented by the blue dashed curve. The displacement ratio of the left side over the right side, L/R is presented as a function of input frequency (black solid curve). Magenta domain indicates the right propagation mode, while the blue domain reveals the left propagation mode. Grey region is the intermediate mode defined in main text.

The beam splitter results continue to be of poor quality, and as far as I can tell from looking at fig 4b, the simulations indicate significantly higher quality results should be obtained. (One presumes this may be due to the effects of added mass.) Moreover, How do any of the experimental results prove a topological mode is observed? I still do not follow their line of reasoning. It seems it should be possible to switch the mode on and off by altering strain -- I guess this is what the video is supposed to show, but by only measuring at a single frequency this allows for the possibility that it has simply changed frequencies. Normally when a paper with both theoretical and experimental results, you measure the same thing in both and show they agree. I do not understand why they have not done this.

We thank the reviewer's comments. Here, we show the reviewer the logical line of this paper. First, we illustrate a topological phase diagram according to the analysis of bulk band topology. Second, we choose two metamaterials with two different topological phases and show the generation of topological states. Third, we observed the elastic topological states numerically and experimentally. Fourth, we demonstrate the dynamically tunable elastic topological states by applying mechanical deformation. The Fig. 4c shows the displacement field at 274 Hz that is the numerical frequency of topological state. The localized displacement field is the typical topological state. Moreover, the Fig. 5e shows the displacement field at 272 Hz, 273 Hz, 274 Hz, 275 Hz and 276 Hz, where the topological state can only be observed at the frequency of 274 Hz. Besides, the displacement fields in Fig. 5d show good agreement with the theory prediction in Fig. 5b. We also show the comparison of experiment results and simulation results in Supplementary Fig. 8, where good agreement can be found. These are strong evidences to prove that we observe the elastic topological states.

As for the Supplementary movie, it seems that the reviewer has understood the key point: we switch the topological mode on and off by applied strain. What we would like to demonstrate is the appearance and disappearance of the interface state when we fix the input frequency and deform the metamaterial, so the input frequency should be a single value.

We added the Supplementary Fig. 8 to show the agreement of numerical and experimental results

explicitly. We also revised the description on Supplementary movie in Supplementary note 2.

The Supplementary Fig. 8 is shown as below:

Supplementary Figure 8 | The comparison of experiment results and simulation results. a, From left to right, the experimentally measured and numerically simulated displacement field distributions at four selected strains and corresponding frequencies. The markers and colors have their correspondences in Fig. 5b. **b.** From left to right, the experimentally measured and numerical simulated displacement field distributions at five selected frequencies at strain of 5.56% along the purple dashed line in Fig. 5b, 272 Hz, 273 Hz, 274 Hz, 275 Hz and 276 Hz.

The revised Supplementary note 2 is shown as below:

“When we deform the metamaterial, the measured acceleration value is fluctuating. The appearance of the topological interface states is observed when the acceleration value reaches the maximum, which is marked in the Supplementary Movie. The topological interface states have the typically enhanced localized field, so the dynamical curve reflects the appearance and disappearance of interface state.”

The revised manuscript in paragraph 3 page 7 is shown as below:

“It is noted that the vibration at the left of the interface drops slower than that at the right, which is consistent with the simulation result in Fig. 4c and the field distribution in Fig. 4b.”

The revised manuscript in paragraph 2 page 9 is shown as below:

“The explicit comparison of experiment results and simulation results is displayed in

Supplementary Fig. 8, where the good agreement of displacement field can be observed.”

The revised manuscript in paragraph 3 page 9 is shown below:

“The experiment details are shown in Supplementary note 2.”

Finally, numerous grammatical errors still exist, and the writing is (still) hard to follow in many places. I would encourage the authors to hire a copy editor.

We thank the reviewer’s comments. We have revised the grammatical errors.

REVIEWERS' COMMENTS:

Reviewer #3 (Remarks to the Author):

Concerning the manuscript, the author addressed all my concerns. The quality of the experimental and numerical results deserves publication in Nature Communications.

My only minor concern is about the movie. I still think the movie is of poor pedagogical quality. As it is not strictly necessary for the article, the author should suppress it or made substantial efforts to improve it (for instance, add some notes, arrows, stops and zooms corresponding to each sequence and improve the background).

Reviewer #3 (Remarks to the Author):

Concerning the manuscript, the author addressed all my concerns. The quality of the experimental and numerical results deserves publication in Nature Communications.

We thank the reviewer's positive comment.

My only minor concern is about the movie. I still think the movie is of poor pedagogical quality. As it is not strictly necessary for the article, the author should suppress it or made substantial efforts to improve it (for instance, add some notes, arrows, stops and zooms corresponding to each sequence and improve the background).

We thank the reviewer's comments. We revised the supplementary movie and made it clear by adding plays, pauses, notes and arrows.